# Structural basis for an early stage of the photosystem II repair cycle in *Chlamydomonas reinhardtii*

Anjie Li [1,2,6], Tingting You [3,6], Xiaojie Pang [2,4], Yidi Wang [1,2], Lijin Tian [2,4], Xiaobo Li [3,5] ✉ & Zhenfeng Liu [1,2] ✉

Photosystem II (PSII) catalyzes water oxidation and plastoquinone reduction by utilizing light energy. It is highly susceptible to photodamage under high-light conditions and the damaged PSII needs to be restored through a process known as the PSII repair cycle. The detailed molecular mechanism underlying the PSII repair process remains mostly elusive. Here, we report biochemical and structural features of a PSII-repair intermediate complex, likely arrested at an early stage of the PSII repair process in the green alga *Chlamydomonas reinhardtii*. The complex contains three protein factors associated with a damaged PSII core, namely Thylakoid Enriched Factor 14 (TEF14), Photosystem II Repair Factor 1 (PRF1), and Photosystem II Repair Factor 2 (PRF2). TEF14, PRF1 and PRF2 may facilitate the release of the manganese-stabilizing protein PsbO, disassembly of peripheral light-harvesting complexes from PSII and blockage of the $Q_B$ site, respectively. Moreover, an α-tocopherol quinone molecule is located adjacent to the heme group of cytochrome $b_{559}$, potentially fulfilling a photoprotective role by preventing the generation of reactive oxygen species.

During oxygenic photosynthesis, photosystem II (PSII) is the primary supramolecular machinery responsible for generating reducing power by utilizing light energy and oxidizing water molecules into oxygen and protons[1]. When excessive excitation accumulates and induces the formation of reactive oxygen species (ROS), the electron transport components within PSII may become impaired due to direct inactivation of the manganese cluster by ultraviolet or visible light radiation and oxidative damages caused by ROS[2,3]. Damages to the PSII core components and the peripheral light-harvesting complex II (LHCII) by the photooxidative stress have been detected at the molecular level in a series of studies[4-7].

To restore the function of PSII upon damage, oxygenic phototrophs have evolved a highly sophisticated process known as the PSII repair cycle[8-10]. The cycle is fairly conserved in various oxygenic photosynthetic organisms from cyanobacteria to higher plants[11]. In green algae and higher plants, the PSII repair cycle may include the following steps[12]: a) dissociation of LHCII from PSII-LHCII supercomplex (PSII-SC), b) degradation of the damaged D1 apoprotein, c) regeneration of the D1 subunit, and d) reassembly of PSII-SC. More than 60 auxiliary factors were reported to be involved in PSII repair cycle and de novo assembly[13], suggesting that the process may involve a highly complex and dynamic system with multiple intermediate states.

[1]Key Laboratory of Biomacromolecules (CAS), National Laboratory of Biomacromolecules, CAS Centre for Excellence in Biomacromolecules, Institute of Biophysics, Chinese Academy of Sciences, Beijing 100101, China. [2]College of Life Sciences, University of Chinese Academy of Sciences, Beijing 101408, China. [3]Key Laboratory of Growth Regulation and Translational Research of Zhejiang Province, School of Life Sciences, Westlake University, Hangzhou 310024, China. [4]Key Laboratory of Photobiology, Institute of Botany, Chinese Academy of Sciences, Beijing 100093, China. [5]Institute of Biology, Westlake Institute for Advanced Study, Hangzhou 310024, China. [6]These authors contributed equally: Anjie Li, Tingting You. ✉e-mail: lixiaobo@westlake.edu.cn; liuzf@ibp.ac.cn

Under high-light conditions, the peripheral antennae of PSII (including LHCII trimers, CP29, CP26, and CP24) may dissociate from the PSII core[14,15]. Such a disassembly process reduces the excitation energy influx to the damaged PSII reaction center[16], and also allows the subsequent PSII core disassembly followed by degradation of D1 protein[17]. Previous studies indicated that the Stn8-induced phosphorylation of PSII core proteins and the Stn7-induced phosphorylation of CP29 may both contribute to inducing the detachment of LHCIIs from PSII under high light[18,19]. Further mechanistic details regarding how LHCII dissociates from PSII remain unclear.

Following the dissociation of peripheral antennae, efficient repair of the damaged PSII also requires further disassembly of PSII dimer (PSII-D) and PSII monomer (PSII-M)[8,10]. It was found that Maintenance of PSII under High light 2 (MPH2), a chloroplast thylakoid lumen protein, stimulates the disassembly of PSII-M and enhances D1 degradation during the PSII repair process[20]. Besides, the release of PsbO from PSII core may promote degradation of photodamaged D1 protein and is thus important for the PSII repair process[21]. The hydrolysis of GTP by PsbO[21] and degradation of free PsbO[22] may facilitate this process. Nevertheless, the PsbO subunit in *Chlamydomonas reinhardtii* does not possess a GTPase domain[23], and the factor/factors involved in PsbO dissociation in the early stage of PSII repair is still an enigma.

The photodamage inactivation and release of the Mn cluster from the oxygen-evolving complex (OEC) is an early event of the PSII photoinhibition process[24], and can cause the formation of hazardous long-lived $P680^{+3}$. Several mechanisms have been discovered to account for quenching of the prolonged $P680^+$, including the direct charge recombination process[25] and the PSII cyclic electron transport[26]. Besides, a PSII-assembly related protein Psb28 was reported to distort the $Q_B$ binding site and modify the coordination environment of the non-heme iron[27], thus switching the PSII core complex into a photoprotective state.

Here, we report biochemical, structural and functional characterization of an intermediate complex of the PSII repair cycle. Three protein factors associated with a damaged PSII monomer, namely Thylakoid Enriched Factor 14 (TEF14), Photosystem II Repair Factor 1 (PRF1) and Photosystem II Repair Factor 2 (PRF2), have been identified and functionally characterized by using knockout mutants generated through the CRISPR-Cas9 method[28]. An α-tocopherol quinone molecule is found in a stromal surface pocket nearby PsbF, potentially function as a photoprotective electron acceptor of cytochrome (Cyt) $b_{559}$ in the PSII repair process.

## Results

### Preparation and characterization of an intermediate complex of the PSII repair process

It is a highly challenging task to capture the intermediate complexes of the PSII repair process, because of their low abundance, heterogeneity, instability and transient existence[8]. We reasoned that the intermediate complex may be enriched in the mutant with a specific PSII-repair-related factor-encoding gene knocked out. Previously, it was reported that dephosphorylation of D1 protein is a prerequisite for proteolytic degradation of the damaged protein[29]. It was also demonstrated that the PSII core phosphatase PBCP is involved in dephosphorylation of PSII core proteins[30], and degradation of the D1 subunit of PSII is diminished in the *pbcp* mutants[31]. Besides, Protein Phosphatase 1 (PPH1) is the other chloroplast protein phosphatase that mainly dephosphorylates LHCII proteins[32,33]. It is also able to dephosphorylate PSII core proteins including D1 when it is overexpressed in *Arabidopsis thaliana*[33] or incubated with thylakoid membrane under in vitro conditions[34]. In *Chlamydomonas reinhardtii*, PBCP and PPH1 may have distinct but overlapping sets of targets, and the presence of PPH1 in the *pbcp* mutant could still dephosphorylate PSII core subunits at a lower rate than PBCP[35]. Therefore, in order to capture the PSII core complexes at an intermediate state of the PSII repair cycle, we cultured the *pph1;pbcp* (a double phosphatase mutant strain[35]) and WT *Chlamydomonas reinhardtii* cells under high light conditions (330 µmol photons $m^{-2} s^{-1}$), and separated the photosynthetic complexes by sucrose density gradient (SDG) ultracentrifugation. The PSII-M fraction appears more abundant in the *pph1;pbcp* strain than the wild type (WT, Fig. 1a). As revealed by SDS-PAGE and mass spectrometry analysis, both *pph1;pbcp* and WT cells contain several PSII-M-associating protein factors under high-light conditions compared to the cells grown under low light (20 µmol photons $m^{-2} s^{-1}$) (Fig. 1b). Among them, a protein with an apparent molecular weight of ~14 kDa was identified as the Thylakoid Enriched Factor 14 (TEF14, Uniprot ID: A8HY43). *Cr*TEF14 was originally found through proteomic and genomic studies[36,37], sharing 31.4% sequence identity with MPH2 from *Arabidopsis thaliana* (*At*MPH2, an ortholog of *Cr*TEF; Fig. 1c). Previously, *At*MPH2 was reported to be involved in the disassembly of monomeric PSII complexes during PSII repair[20]. Therefore, the PSII-M complex we obtained likely represents an intermediate state of the PSII repair process in green algae and plants. Two other smaller proteins with apparent molecular weights between 6.5 and 14.4 kDa were identified as the products of *Cre03.g164300* and *Cre01.g051500/Cre01.g042200*, and are hereby named as Photosystem II Repair Factor 1 (PRF1, Uniprot ID: A0A2K3DWM2) and Photosystem II Repair Factor 2 (PRF2, Uniprot ID: A8HNG8/PRF2′, Uniprot ID: A8HMM7) respectively.

The levels of TEF14, PRF1, and PRF2/PRF2′ (relative to a major core subunit of PSII, namely CP47) proteins in the PSII-TPP complex are evidently increased when compared to the corresponding ones in the PSII-M sample from the low-light adapted cells (Supplementary Fig. 1a). Further quantitative reverse transcription polymerase chain reaction (RT-qPCR) analysis results indicate that the transcription levels of *TEF14* and *PRF2* are increased and reach plateaus after 4 h of high-light treatment, whereas the transcription level of *PRF1* is not as sensitive to high-light treatment as TEF14 or PRF2 (Supplementary Fig. 1b–d and Table 5). In addition, TEF14 and PRF2 contain multiple phosphorylation sites located with high confidence[38], suggesting that these two proteins may be regulated at both transcriptional and post-translational levels.

### Architecture and assembly mechanism of the PSII repair intermediate complex PSII-TPP

To understand the functional roles of TEF14, PRF1 and PRF2 in PSII repair, the structure of the PSII-M-TEF14-PRF1-PRF2 (PSII-TPP) complex obtained from the PSII-M fraction of the high-light adapted *pph1;pbcp C. reinhardtii* cells has been solved at 2.6 Å resolution through the single-particle cryo-EM method (Supplementary Fig. 2 and Table 1). The complex is composed of a partly damaged PSII core monomer and includes all three identified protein subunits, namely TEF14, PRF1 and PRF2 (Fig. 2a). The identities of the three protein factors were verified by the consistency between the cryo-EM densities and the amino acid sequences of the proteins identified through mass-spectrometry (Fig. 1b). Meanwhile, the PSII core region contains four major PSII core subunits (D1, D2, CP43, CP47) and 11 small membrane-intrinsic proteins (PsbE, PsbF, PsbI, PsbH, PsbK, PsbL, PsbM, PsbTc, PsbX, PsbZ and Ycf12), whereas the peripheral antenna complexes (LHCII, CP29 or CP26), PsbJ, PsbR, PsbW and three extrinsic subunits on lumenal surface (PsbO, PsbP and PsbQ) were all absent in the complex.

Notably, the three extrinsic proteins of the oxygen-evolving complex and the Mn-cluster responsible for water splitting are all missing according to the local features in the cryo-EM density map (Supplementary Fig. 3a). In addition, the DE-loop between transmembrane helices D and E of D1 (E226-T245), which is a candidate site for D1 degradation[39], is not observed in the cryo-EM map either (Supplementary Fig. 3b). Further western blot (WB) analysis against the carboxy-terminal and amino-terminal regions of the D1 apoprotein reveals that the high-light adapted PSII-TPP sample contains two

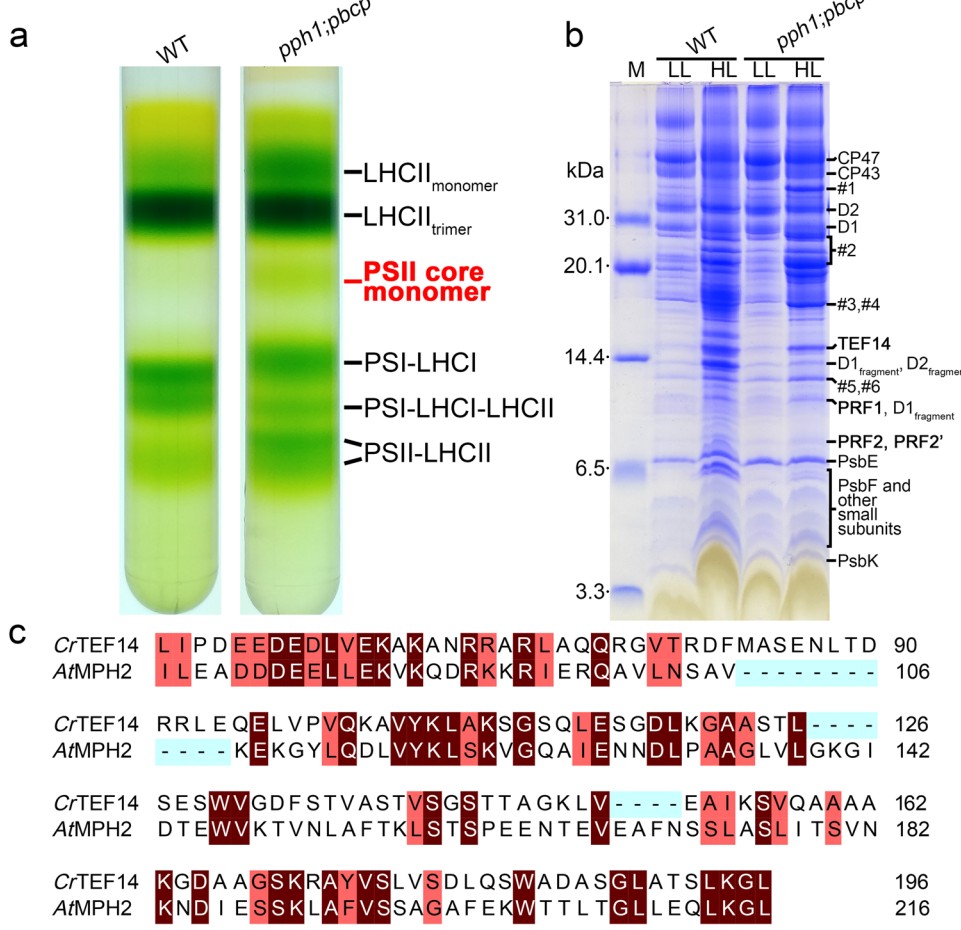

**Fig. 1 | Sample preparation and characterization of a PSII-repair complex.**
**a** Separation of *C. reinhardtii* photosynthetic complexes through the sucrose density gradient (SDG) ultracentrifugation method. The photosynthetic complexes were extracted from the high-light acclimated WT and *pph1;pbcp C. reinhardtii* cells with α-DDM. **b** Protein composition analysis of the PSII core monomer (PSII-M) fractions from the SDG tubes by SDS-PAGE. The comparison of SDS-PAGE bands between the control (LL, low-light adapted PSII-M) and the high-light adapted PSII-M samples (HL) led to the discovery of four proteins in the sample, namely TEF14, PRF1, PRF2, and PRF2'. Their identities were revealed through mass spectrometry on the tryptic digestion products of the corresponding protein bands excised from

the gel. The bands labeled as #1-6 contain various contaminant proteins as identified by mass spectrometry, namely Cyt *f* (#1), LhcbM proteins (#2), Lhca1 (#3), Cyt $b_6$ (#4), Psb27 (#5) and Lhca7 (#6). The image is a representative of three independent repeats with similar results. Source data are provided as a Source Data file. **c** TEF14 from *C. reinhardtii* is an ortholog of *At*MPH2 as revealed by the sequence alignment result. The two protein sequences excluding the transit peptides share an identity of 31.4%. The transit peptide regions are omitted for clarity. Identical residues, similar residues and absent ones are in dark red, light red, and cyan, respectively.

proteolytic fragments of D1, namely a ~11 kDa carboxy-proximal fragment and a ~18 kDa amino-proximal fragment (Supplementary Fig. 3c-e). Such a degradation pattern corresponds to a previously reported cleavage site located in the DE-loop of D1 (ref. 40 and Supplementary Fig. 3f). After proteolytic cleavage at the DE-loop region, the spatial restraint of the loop may be reduced so that it becomes too flexible to be observed in the cryo-EM map. Therefore, the PSII-TPP complex might represent an intermediate state at an early stage of the PSII repair cycle with a partially-degraded D1 subunit.

### TEF14 facilitates PsbO release and D1 degradation

TEF14 is a soluble protein binding on the lumenal surface of CP47 (Fig. 2c). The amino-terminal region (NTR, Glu55−Glu86) of TEF14 is curved by ~90° at Asn67 to wrap around the lumenal domain of CP47, serving as an adapter for TEF14 to associate with the PSII core (Supplementary Fig. 4a). The acidic residue-rich "$D^{54}EEDE^{58}$" motif in the NTR_{TEF14} forms electrostatic interactions with the basic residues of CP47 (Lys419, Arg422, Lys423 and Arg434, Supplementary Fig. 4b and c). The downstream "$K^{65}$xxRRARxxxQ$^{75}$" motif in the NTR_{TEF14} also forms electrostatic interactions and hydrogen bonds with the lumenal

residues from CP47 (Supplementary Fig. 4e and f). Moreover, Phe82 of TEF14 is embedded in a hydrophobic pocket at the lumenal surface of CP47 (Supplementary Fig. 4d). The subsequent five α-helix bundle (FAB) domain of TEF14 is also involved in binding CP47, but the interactions are weaker than that of NTR_{TEF14}-CP47. As a result, the FAB domain may adopt variable conformations relative to the NTR of TEF14 and the PSII core (Supplementary Movie 1).

Similar to the *mph2* plant (previously reported to be deficient in PSII repair)[20], the *tef14* mutant exhibited greater loss of PSII maximum quantum efficiency than the WT cells under high light stress (Fig. 3a). When the structure of PSII-TPP complex is superposed with the low-light adapted PSII-LHCII supercomplex from *C. reinhardtii*[41] (PDB ID: 6KAC), it is evident that the binding site of TEF14 is in steric hindrance with that of PsbO at multiple sites (Fig. 3b). The amino-terminal loop of TEF14 (I52-E56) clashes with the loop (N216-E232) of PsbO within the same PSII monomer of the $C_2S_2$ dimer. Moreover, the amino-terminal helix (R68-A73) and the carboxy-terminal α-helical bundle of TEF14 clash with the loop (V106-T122) and the β-barrel domain of PsbO' from the adjacent PSII monomer. These observations suggest that binding of TEF14 to CP47 may induce dissociation of PsbO from PSII core, or it

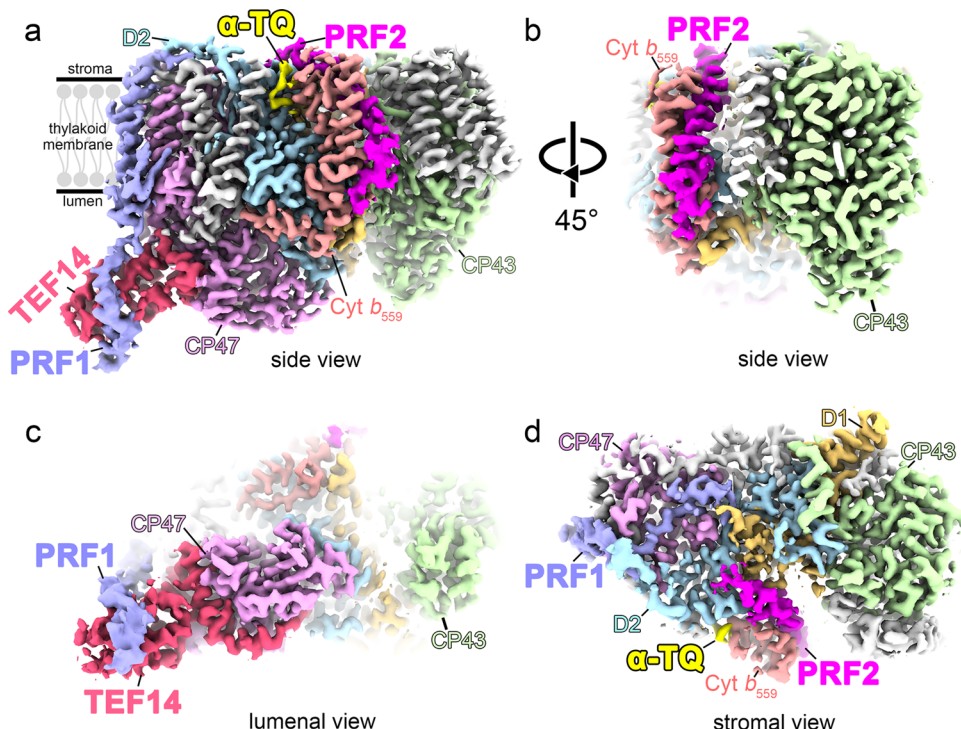

**Fig. 2 | Cryo-EM map of the PSII-TPP complex involved in the PSII repair process. a** Overall architecture of the PSII-TPP complex showing the binding sites of TEF14, PRF1 and PRF2. TEF14, PRF1, PRF2 and an α-tocopherol quinone molecule are colored in red, light purple-blue, magenta and yellow, respectively. D1, D2, CP47, CP43 and Cyt $b_{559}$ are colored gold, cyan, pink, green and salmon, respectively. The remaining small membrane protein subunits are colored in gray. **b** A rotated side view of the PSII-TPP complex showing the PRF2 binding site near Cyt $b_{559}$. **c** Bottom view of the complex from the thylakoid lumenal side showing the binding site of TEF14 and the carboxy-terminal region of PRF1. **d**, Top view from the chloroplast stromal side showing the binding site of PRF2 and the amino-terminal region of PRF1.

may block the re-association of PsbO which was released by other factor(s). As the PSII dimer interface on the lumenal side is mainly stabilized by the interactions between the Val106-Thr122 loop region of PsbO and a lumenal loop (Phe45-Ser92) of CP47 (Fig. 3b), dissociation of PsbO may further destabilize the PSII dimer and facilitate its disassembly into PSII monomers.

To investigate the function of TEF14, we analyzed the PsbO-binding pattern and the PSII complex fractions from the high-light (330 µmol photons·m$^2$·s$^{-1}$) adapted WT and *tef14* (constructed by using the CRISPR-Cas9 method, see methods and supplementary Fig. 11 and Tables 2–4 for details) cells. In the WT cells, TEF14 mainly migrates in the PSII-M band region, whereas only a small fraction of it is detected in the PSII-SC fraction (Fig. 3c). In the *tef14* mutant, the abundance of PsbO in the PSII-D sample is increased by more than one-fold in comparison with the one from WT, whereas the level of PsbO in the PSII-M sample from the *tef14* mutant is similar to that of WT (Fig. 3d). Moreover, an extra green band corresponding to PSII-D appeared between the PSII-M and PSI-LHCI bands in the *tef14* cells when compared with the WT (Fig. 3e). The results suggest that dissociation of PsbO from PSII-D and the disassembly of PSII-D into PSII-M are likely deficient in the absence of TEF14.

To test whether TEF14 is directly involved in dissociating PsbO from PSII or not, we purified the recombinant TEF14 protein expressed in *E. coli*, and incubated it with the solubilized thylakoid sample from the *tef14* mutant (Fig. 3f). The relative amount of PsbO in the PSII-D sample decreased slightly upon the addition of recombinant TEF14 protein (relative to the one without the exogenous TEF14 protein), whereas there was no significant difference in the PSII-M samples with and without TEF14 added (Fig. 3g). The result suggests that while TEF14 may facilitate the dissociation of PsbO from PSII-D, the protein alone is not sufficient to fully disassemble PsbO from PSII-D. It may cooperate with some other factors in vivo, such as a lumenal protease responsible

for the degradation of PsbO as reported previously[22], to release PsbO from PSII-D.

Previously, it was reported that mutation of *At*MPH2 caused inhibition of D1 degradation during the PSII repair cycle[20]. To investigate the role of TEF14 in *C. reinhardtii* PSII repair, we analyzed D1 degradation kinetics in the *tef14* mutant. Under continuous high light illumination and with lincomycin added to the cells to block de novo synthesis of chloroplast-encoded proteins, D1 degradation in *tef14* cells exhibited a lower rate than that of the WT cells. The difference became more evident after 4 h, when the relative D1 amount dropped to only 52% in WT cells but remained at 80% in *tef14* cells (Fig. 3h). In the low light-adapted PSII supercomplexes, PsbO covers the lumenal loops of D1 protein and protects them from proteolytic degradation[41]. It was reported that the extent and kinetics of PsbO dissociation correlate with those of D1 degradation in both spinach and *C. reinhardtii*[21,42]. As TEF14 may function to facilitate dissociation of PsbO from PSII-D, it may also render the lumenal loops of D1 protein accessible to the lumenal proteases, such as DEG1, DEG5 and DEG8.

**PRF1 is involved in dissociation of peripheral antennae from PSII**
PRF1 is a small membrane protein with one long membrane-spanning α-helix located adjacent to CP47 (Fig. 2a, c, d). The amino-terminal domain of PRF1 forms an irregular hairpin-like structure at the stromal side, interacting closely with five PSII core subunits, namely CP47, D1, D2, PsbH and PsbL, through 34 hydrogen bonds and two salt bridges (Supplementary Fig. 5a–d). The upstream region of the hairpin structure (Asp35-Phe55) docks into a surface pocket formed by D1, D2, CP47 and PsbL (Supplementary Fig. 5a, b). The downstream region (Pro56-Ala61) of the hairpin domain in PRF1 forms a straight strand settling in a groove on the stromal surface formed by CP47 and PsbL (Supplementary Fig. 5c). Two subsequent Tyr residues (Tyr62, Tyr63) orient

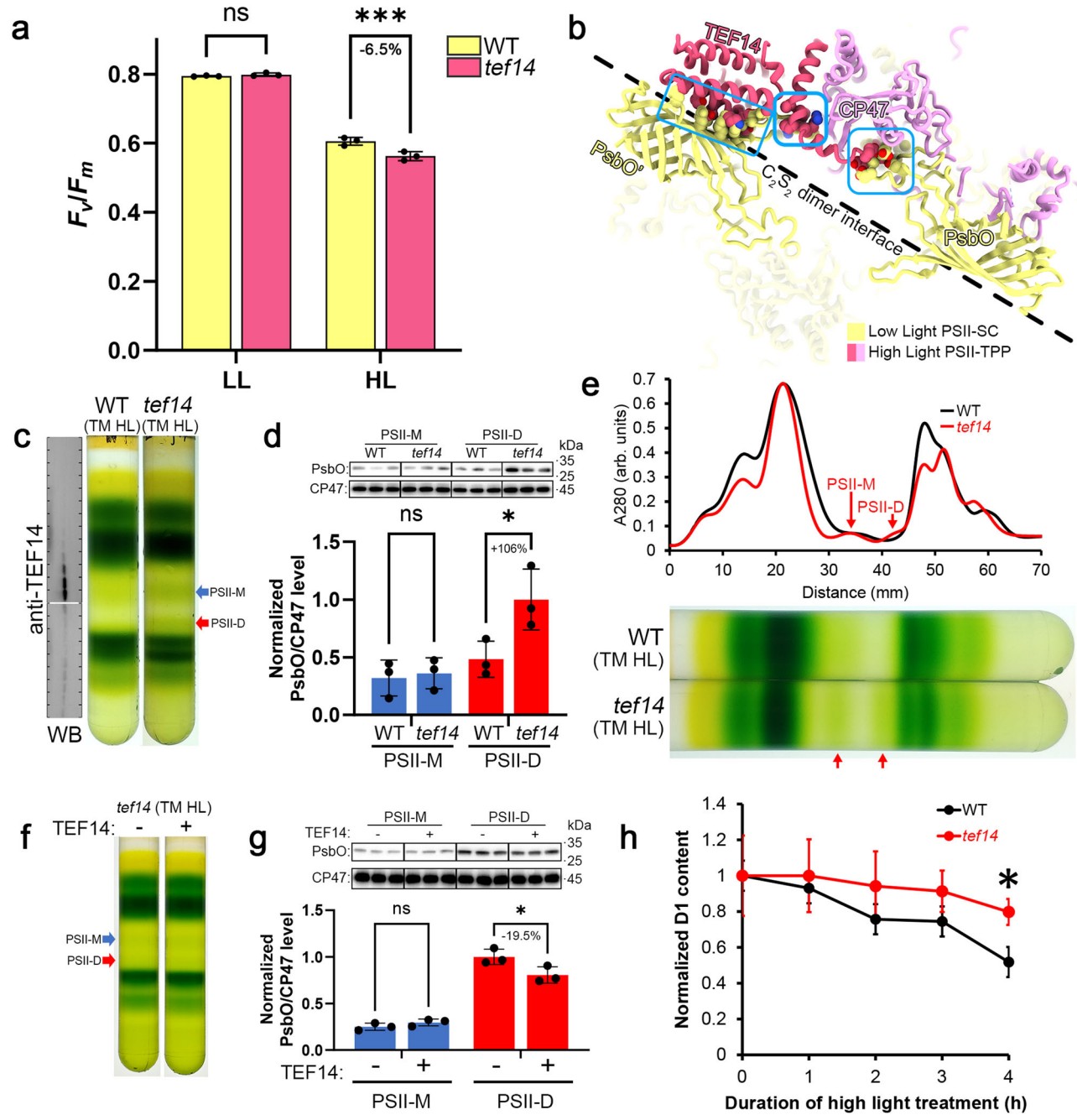

their bulky side chains towards opposite directions, and are sandwiched between PsbH and the amino-terminal region of D2 (Supplementary Fig. 5d). The transmembrane helix of PRF1 interacts with the helix III and helix IV of CP47 via van der Waals interactions. The carboxy-terminal domain of PRF1 extends to the lumenal surface, wrapping around the α-helices (III and VII) of TEF14 (Supplementary Fig. 5e).

Remarkably, the binding sites of PRF1 in the PSII-TPP complex are mainly on the stromal surface and membrane-embedded surface of CP47. They overlap largely with the binding sites for CP29 in the low light adapted PSII-SC (Fig. 4a). The amino-terminal region of PRF1 occupies the same stromal-surface binding site for the long hairpin loop at the amino-proximal region of CP29, whereas the transmembrane helix of PRF1 takes up the binding site for Chl $a$603, Chl $a$609 and Chl $b$607 in the membrane-embedded domain of CP29 (Supplementary Fig. 6a, b). Therefore, binding of PRF1 to CP47 will most likely

prevent CP29 from binding to CP47 of the PSII core due to steric hindrance.

To further investigate the functional role of PRF1, we generated the $prf1$ mutant strain of $C.\ reinhardtii$. After high-light treatment for 8 h, the $F_v/F_m$ of the $prf1$ cells is much lower than the wild type (Fig. 4b). The relative amount of PSII-SC ($vs$ the PSI-LHCI fraction) in the $prf1$ cells is evidently much higher than the corresponding fraction in the WT cells according to the normalized absorption at 280 nm wavelength (Fig. 4c and Supplementary Fig. 6c). In addition, the amount of monomeric (CP29 and CP26) and trimeric LHCIIs appears lower than that of the WT (Fig. 4c). Therefore, PRF1 may participate in promoting disassembly of PSII-SC into PSII and free LHCIIs. We further analyzed the assembly of PSII-SCs in the $prf1$ cells through the negative-staining electron microscopy and 2D classifications. Curiously, the amount of $C_2S$ form, which lacks the peripheral antennae on one side, was increased from 1.9% (WT) to 18% in the $prf1$ mutant (Fig. 4e). At least in

**Fig. 3 | TEF14 functions in facilitating the dissociation of PsbO during PSII repair process. a** The maximal quantum efficiency ($F_v/F_m$) of PSII from the WT and *tef14* cells treated with regular growth light (40 μE) and high light (750 μE). The $F_v/F_m$ values are presented as mean ± SD of three technical replicates. The significance of difference was analyzed by using the two-way ANOVA with two-sided Šidák's multiple comparisons correction ($P = 0.81$ for low-light/LL WT and *tef14* samples, $P = 0.0008$ for high-light/HL samples; ns, not significant, $P > 0.05$; ***, $P < 0.001$). **b** Structure of the PSII-TPP is superposed with that of the intact PSII-LHCII supercomplex (PDB: 6KAC, aligned on the CP47 subunits). The clashing regions between TEF14 and PsbO are shown as spheres, and framed by blue rectangles. **c** SDG ultracentrifugation of the detergent-solubilized thylakoid membrane (TM) prepared from the high-light (HL) adapted WT and *tef14* cells. The western blot (WB) analysis with antibodies against TEF14 is shown on the right side. **d** Western blot analysis of PsbO and CP47 for estimating their relative abundance in the PSII-M and PSII-D in WT and *tef14*. The blot intensity ratio between PsbO and CP47 (PsbO/CP47) are represented as mean ± SD of three technical replicates, and the highest PsbO/CP47 value was normalized to 1 for comparison. The significance of difference was analyzed by using the two-way ANOVA with two-sided Šidák's multiple comparisons correction ($P = 0.96$ for the WT and *tef14* PSII-M samples, $P = 0.018$ for the WT and *tef14* PSII-D samples; ns, not significant, $P > 0.05$; *, $0.01 < P \le 0.05$). **e** Comparison of the photosynthetic complex content from the

WT and *tef14* thylakoids prepared as in (**c**). The red arrows indicate the over-accumulated PSII-M and PSII-D (PSII core dimer) fractions in the *tef14* thylakoid. The UV280 absorption profiles of the SDG fractions were scanned, normalized on the largest peaks of LHCII trimer fraction, and aligned with the positions of the SDG bands. **f** The SDG ultracentrifugation analysis on the detergent-solubilized fractions of the high-light adapted *tef14* thylakoid membrane incubated with or without the addition of the purified TEF14 protein. **g** Western blot analysis of PsbO and CP47 in the PSII-M and PSII-D from the SDG fractions shown in (**f**). The blot intensity ratio between PsbO and CP47 (PsbO/CP47) was represented as mean ± SD of three technical replicates, and the highest PsbO/CP47 value was normalized to 1 for comparison. The significance of the difference was analyzed by using two-way ANOVA with two-sided Šidák's multiple comparisons correction ($P = 0.66$ for the PSII-M samples with and without TEF14 added, $P = 0.013$ for the PSII-D samples with and without TEF14 added; ns, not significant, $P > 0.05$; *, $0.01 < P \le 0.05$). **h** The degradation kinetics of D1 protein under high light (330 μmol photons $m^2 s^{-1}$) with 1 mM lincomycin. The amount of D1 protein was quantified using the western blots probed with antibody against D1 amino-terminal sequences. The amount of D1 for each time point was normalized to the one before the onset of high light illumination (0 h). The values are represented as mean ± SD of three technical replicates. Significance of differences was analyzed using the student's *t* test ($P = 0.027$; *, $0.01 < p \le 0.05$). Source data are provided as a Source Data file.

part, the $C_2S$ might belong to an intermediate form of the de novo assembly of the PSII-SC, in addition to the partially disassembled form of PSII-SC during the PSII repair process. The abundant accumulation of the intact $C_2S_2$ supercomplexes in both WT and *prf1* strains suggests that the assembly of S-LHCII, CP26 and (most importantly) CP29 with PSII-core is normal in the *prf1* mutant. Besides, it may also suggest that while the overall abundance of PSII-SC ($C_2S$ and larger complexes) has increased in the *prf1* mutant, the stability of intact PSII-SC ($C_2S_2$ and larger complexes) might be decreased, probably due to accumulated photodamaged sites. Previously, Kale et al. have identified multiple ROS modification sites in LHCII (ref. [7]), including Pro164 from Lhcb1 located at the LHCII-CP29 and/or LHCII-CP26 interface. While the antennae dissociation mechanism is blocked in the *prf1* mutant, the excess excitation energy may lead to accumulated oxidative modifications of LHCII besides the PSII core subunits. Taken together, we propose that binding of PRF1 to the exposed PSII-core may prevent the re-association of CP29 to PSII, thus shifting the association-dissociation equilibrium of LHCIIs and PSII towards the dissociated forms. Thereby, association of PRF1 with the PSII core subunits likely prepares the PSII-core complex for further disassembly processes, and helps to alleviate the accumulation of harmful excess excitation energy by preventing re-association of peripheral antenna complexes during the PSII repair process.

The isothermal titration calorimetry (ITC) analysis result indicates that the carboxy-terminal region of PRF1 binds tightly with TEF14 with a dissociation constant ($K_D$) of 215 nM (Supplementary Fig. 5f). Their interactions stabilize the FAB domain of TEF14 by restricting its movement. The distribution pattern of TEF14 among the photosynthetic complexes from the *prf1* cells is essentially the same as the one in the WT cells (Fig. 4c), indicating that the absence of PRF1 does not affect the binding of TEF14 to PSII-M and PSII-D. On the other hand, the absence of PRF1 affects the dissociation of PsbO from PSII. Under high-light conditions, the PSII-M and PSII-D complexes accumulate nearly twice the amount of PsbO in the *prf1* mutant than that in the WT (Fig. 4d). The results indicate that PRF1 and TEF14 may have a synergistic action in releasing PsbO from PSII-M/D, as PRF1 may serve to stabilize TEF14 in a position (Fig. 2a) favorable for stimulating PsbO dissociation.

**PRF2 modulates the lateral gate of $Q_B$ site**

PRF2 (uniport ID: A8HNG8) is the other small single-transmembrane-helix protein associated with the damaged PSII-M. While PsbJ is absent in the PSII-TPP complex, PRF2 takes a position adjacent to that of PsbJ on one side of PsbE and PsbF. In the 20-Å wide cleft between Cyt $b_{559}$

and Ycf12, the transmembrane helix of PRF2 attaches to a side groove between PsbE and PsbF (Fig. 2b), forming van der Waals interactions with them (Supplementary Fig. 7a, b). The carboxy-terminal region is located at the stromal side and clamped by D1, D2, and PsbE (Supplementary Fig. 7c), while the amino-terminal region of PRF2 anchors at a lumenal surface site between PsbE and PsbF (Supplementary Fig. 7d).

The *prf2* mutant cells exhibit an evident decline of quantum efficiency induced by high-light treatment when compared to the WT cells (Fig. 5a). Therefore, PRF2 is likely involved in the maintenance of PSII activity under high-light conditions. The carboxy-terminal region of PRF2 is in close interaction with the DE-loop region (Gln261-Asn267) of D1 (Fig. 5b). In the low-light adapted $C_2S_2$ PSII-SC structure[41], the side chain of Tyr262 in the DE-loop of D1 protein extends towards Cyt $b_{559}$. Such a conformation is in steric hindrance with the Ile32 residue of PRF2 in the PSII-TPP complex obtained from the cells grown under high light. In the PSII-TPP complex, the Gln261-Asn267 loop region of D1 protein adopts a flipped conformation to avoid such steric hindrance, in which $Tyr262_{D1}$ turns towards the gap formed by PRF2, D1 and D2 (Fig. 5b). The dramatic shift of this loop causes the movement of Ser264 and Phe265 away from the pocket of $Q_B$ site for hosting plastoquinone (PQ) head group (Fig. 5c). In addition, the shift of the Gln261-Asn267 loop region further causes the obstruction of the potential PQ entrance, preventing the PQ molecule in the external pool from accessing the $Q_B$ site. Due to a flip-down movement of Gln261 side chain and a rotation of Phe265 side chain, the maximal gate width of the PQ entrance to the $Q_B$ site shrinks to 4.5 Å (Fig. 5d), whereas the gate width is 6.7 Å in the low-light adapted PSII structure (Fig. 5e). A PQ molecule has a minimal van der Waals diameter of 7.7 Å (Fig. 5f), so it is unlikely for PQ to enter the $Q_B$ site through the obstructed gate with a width much smaller than that of PQ. Thus, binding of PRF2 to the PsbE, PsbF and D1 induces blockage of the $Q_B$ site from exchanging $PQH_2/PQ$ with the external pool.

Curiously, we found a lipid-like density at the $Q_B$ site fits poorly with the model of a PQ molecule but resembles a phospholipid, such as a phosphatidylglycerol molecule (Supplementary Fig. 7e). Thereby, the obstruction of the $Q_B$ entrance and the binding of a non-electron-carrier lipid-like molecule at the $Q_B$ site may collectively shut down the electron export from the PSII. A previous study suggested that release of Mn cluster may change the redox potential of $Q_A/Q_A^-$, and when the $Q_B$ site is simultaneously blocked by 3-(3,4-dichlorophenyl)-1,1-dimethylurea, photoprotective charge recombination between $Q_A^-$ and $Tyr_Z^+$ is promoted[27]. Likewise, PRF2 may function to induce blockage on the acceptor side of the PSII-repair intermediate complex and

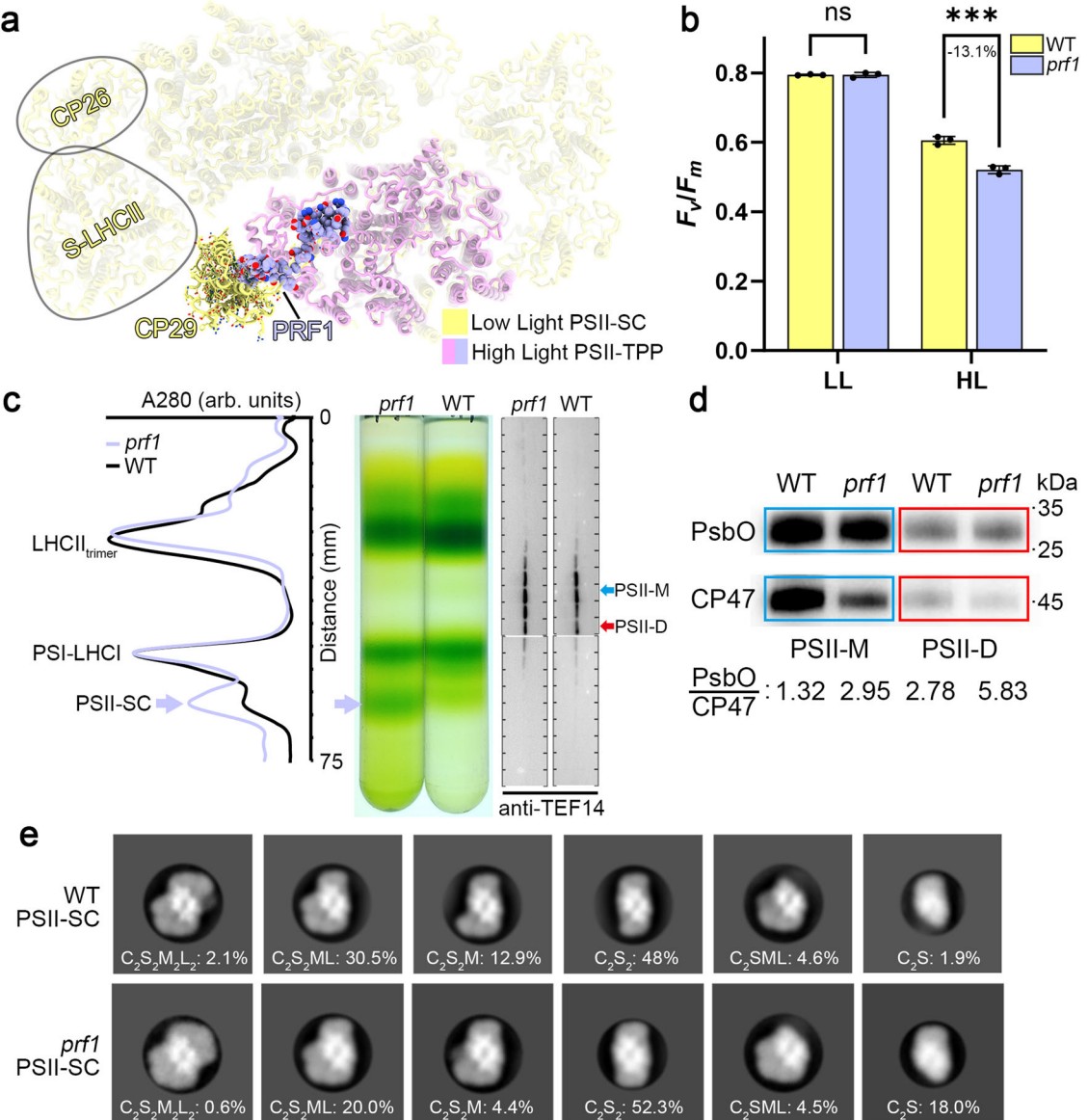

**Fig. 4 | The binding site and function of PRF1. a** Structure of the PSII-TPP complex superposed with that of the low-light adapted PSII-LHCII supercomplex (PDB: 6KAC, aligned on the CP47 subunits). PRF1 in the PSII-TPP is shown as spheres, while CP29 in the PSII-LHCII supercomplex is shown as sticks. **b** The maximal quantum efficiency of PSII from the WT and *prf1* cells adapted to the growth light (LL) and high light (HL) treatment. The values are represented as mean ± SD of three technical replicates. The significance of the difference was analyzed by using the two-way ANOVA with two-sided Šidák's multiple comparisons correction ($P > 0.99$ for the low-light/LL samples, $P < 0.001$ for the high-light/HL samples; ns, not significant, $P > 0.05$; ***, $P < 0.001$). **c** The SDG ultracentrifugation analyses on the solubilized thylakoid samples prepared from the high-light treated WT and *prf1* cells. The UV280 absorption ($A_{280}$) profiles were obtained by measuring $A_{280}$ of the SDG fractions with an automatic fractionator. The bands corresponding to the PSII-LHCII supercomplexes are marked as light purple arrows. The western blot analysis of the TEF14 protein in the SDG fractions is shown on the right. **d** The western blot analyses of the PSII-M/D fraction collected from (**c**), using the antibodies against PsbO and CP47. The relative amount of PsbO is calculated as the grayscale value of PsbO bands divided by that of CP47 and the ratio is presented below. The images are representative of three biological replicates. **e** 2D classifications of the negatively stained particles from the PSII-SC fractions as shown in (**c**). While similar types of PSII-LHCII supercomplexes are present, the percentage of $C_2S_2$ and $C_2S$-type particles is higher in *prf1* than in WT. Source data are provided as a Source Data file.

switch it to a photoprotective state at the early stage of PSII repair. Such a state might help the PSII complex to cope with the redox stress caused by the loss of the Mn-cluster, which is a primary step of photoinhibition[43].

In the high-light treated *prf2* mutant cells, the PSII-M complex contains a small single-transmembrane-helix protein binding to Cyt $b_{559}$ in a way resembling that of PRF2 as revealed by the cryo-EM structure (Supplementary Fig. 8a and 12). The mass spectrometry results for the PRF2-containing SDS-PAGE band (Fig. 1b and Supplementary Fig. 8b) suggest a candidate with the Uniprot ID of A8HMM7.

Its structural model fits well with the cryo-EM density of the additionally-observed protein occupying the PRF2-binding site (Supplementary Fig. 8c). It is named as PRF2', because the protein shares similar amino acid sequences, overall structures and binding sites with those of PRF2. Especially, the membrane-spanning regions of PRF2 and PRF2' are highly similar (Supplementary Fig. 8d). On the other hand, the cryo-EM density of PRF2' is slightly different from that of PRF2, especially at the site interacting with $W35_{PsbE}$ (Supplementary Fig. 8e). When PRF2' is associated with the PSII core (as in Supplementary Fig. 8f), it induces conformational changes on the DE-loop region of D1

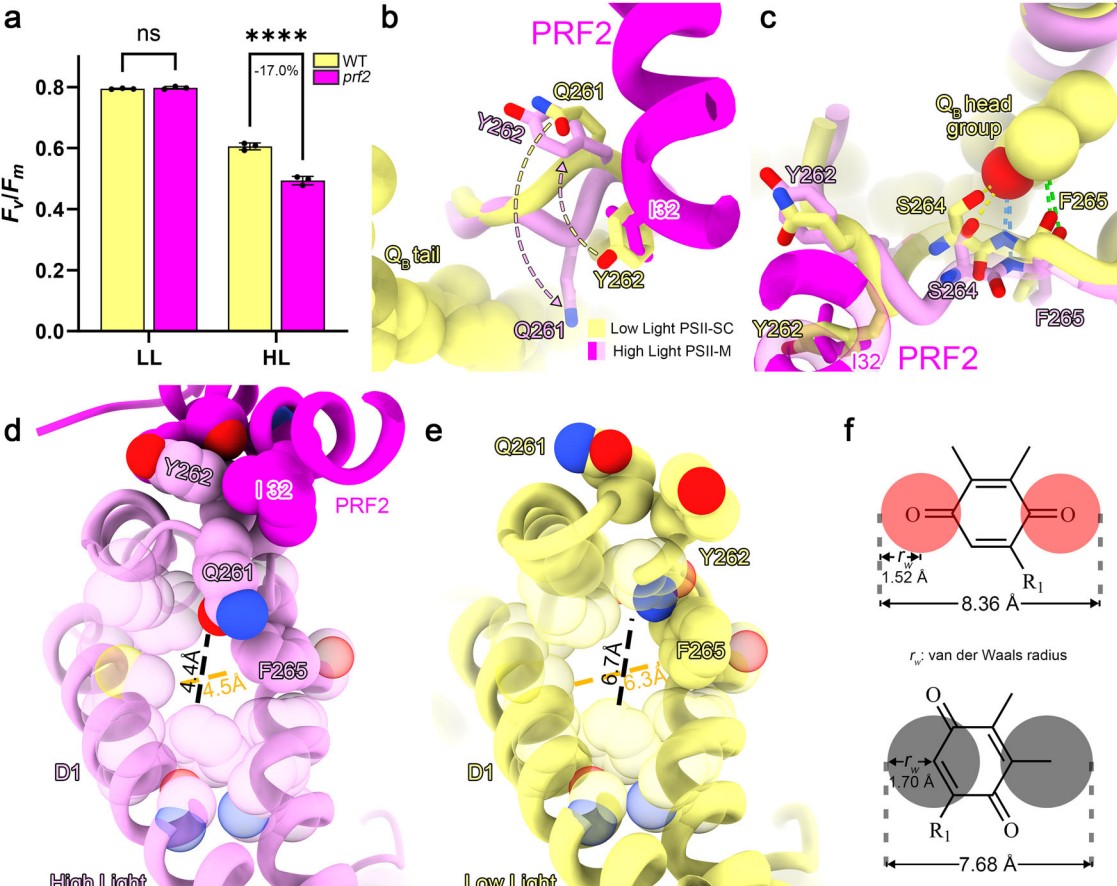

**Fig. 5 | The function of PRF2 by inducing the blockage to the $Q_B$ site of PSII.**
**a** The maximal quantum efficiency of PSII from the WT and *prf2* cells adapted to the growth light (LL) and high light (HL) conditions. The values are represented as mean ± SD of three technical replicates. The significance of difference was analyzed by using the two-way ANOVA with two-sided Šidák's multiple comparisons correction ($P = 0.91$ for the low-light/LL samples, $P < 0.0001$ for the high-light/HL samples; ns, not significant, $P > 0.05$; ****, $P < 0.0001$). Source data are provided as a Source Data file. **b, c** The change of $Q_B$ sites in the PSII-TPP complex (magenta and pink) as shown by superposing it with the low-light adapted PSII-LHCII supercomplex (PDB: 6KAC, yellow, aligned on the D1 subunits). The $Q_B$ molecule from PDB: 3WU2 is docked into the low light PSII-SC model by aligning them on the D1 subunits. The arrows in (**b**) refer to the directions of conformational changes of Q261 and Y262 upon binding of PRF2. **d, e** The $Q_B$ entrances in the high-light adapted PSII-TPP complex (**d**) and the low-light adapted PSII-LHCII supercomplex (**e**). The gate widths are calculated as the distances between two gate-lining atoms minus their van der Waals radius and labeled near the dash lines. **f** The longest and shortest van der Waals diameters of the $Q_B$ head group.

protein in a way similar to the one with PRF2 associated (Supplementary Fig. 8g). Taken together, we suggest that PRF2' may be a redundant homolog of PRF2 functioning in a similar way to inhibit electron transport on the acceptor side of PSII during the PSII repair process. Notably, a protein with the same sequence as PRF2', but named differently as putatively Photosystem B Associated 1, has recently been found to associate mainly with PSII monomer[44], consistent with our findings.

**An α-tocopherol quinone molecule is located nearby Cyt $b_{559}$**

Earlier biochemical studies indicated that some short-chain prenyllipids (including tocopherols and tocopherol quinones) can bind to PSII, and they may have photoprotective roles under high light[45]. However, the binding sites of these small molecules are elusive, and the detailed mechanism of their photoprotective function remains unclear. A recent study reported that an α-tocopherol (α-Toc) molecule binds at the boundary between CP43 and LHCII trimer, and could potentially act as a sensor of excessive irradiation and induce conformational changes to the PSII-LHCII supercomplex[46]. In our PSII-TPP complex, a small molecule is accommodated in a shallow membrane-embedded groove between PsbX and PsbF and near the membrane-water interface on the stromal side (Fig. 6a). The cryo-EM density

matches well with an α-tocopherol quinone (α-TQ) molecule (Fig. 6b). Consistently, the presence of α-TQ in the purified PSII-TPP fraction is verified through the liquid chromatography-mass spectrometry (LC-MS). The precursor ion with a m/z of 429 yielded a signal intensity significantly higher than noises, and its fragment ion spectrum possesses a characteristic main peak with a m/z of 165 (Fig. 6c, d), both consistent with the previous report[47]. The identity of the compound with a precursor ion m/z of 429 is also verified by its LC retention time matching with the standard α-TQ sample (Fig. 6e). The absorption spectrum of α-TQ from the PSII-TPP extract separated by HPLC exhibits a broad characteristic peak ranging from 262 to 268 nm, consistent with that of the standard α-TQ sample (Fig. 6f) and a previous study[48]. We also observed that the levels of α-TQ in the PSII-TPP and the high-light PSII-LHCII supercomplex samples are similar (0.2–0.3 α-TQ per PSII-M) and higher than that in the low-light PSII-LHCII supercomplex sample (<0.1 α-TQ per PSII-M, Fig. 6g; Methods).

The binding site of α-TQ is near the heme group in Cyt $b_{559}$ with a minimal distance of 13.6 Å between their conjugate rings (Fig. 7c). Such a short distance is favorable for electron tunneling to occur. Besides, the closest distance between the carboxyl group of Asp97 from PsbX and the carbonyl group of the α-TQ head group is 2.8 Å (Fig. 6h), which may form a hydrogen bond upon Asp97 protonation. The

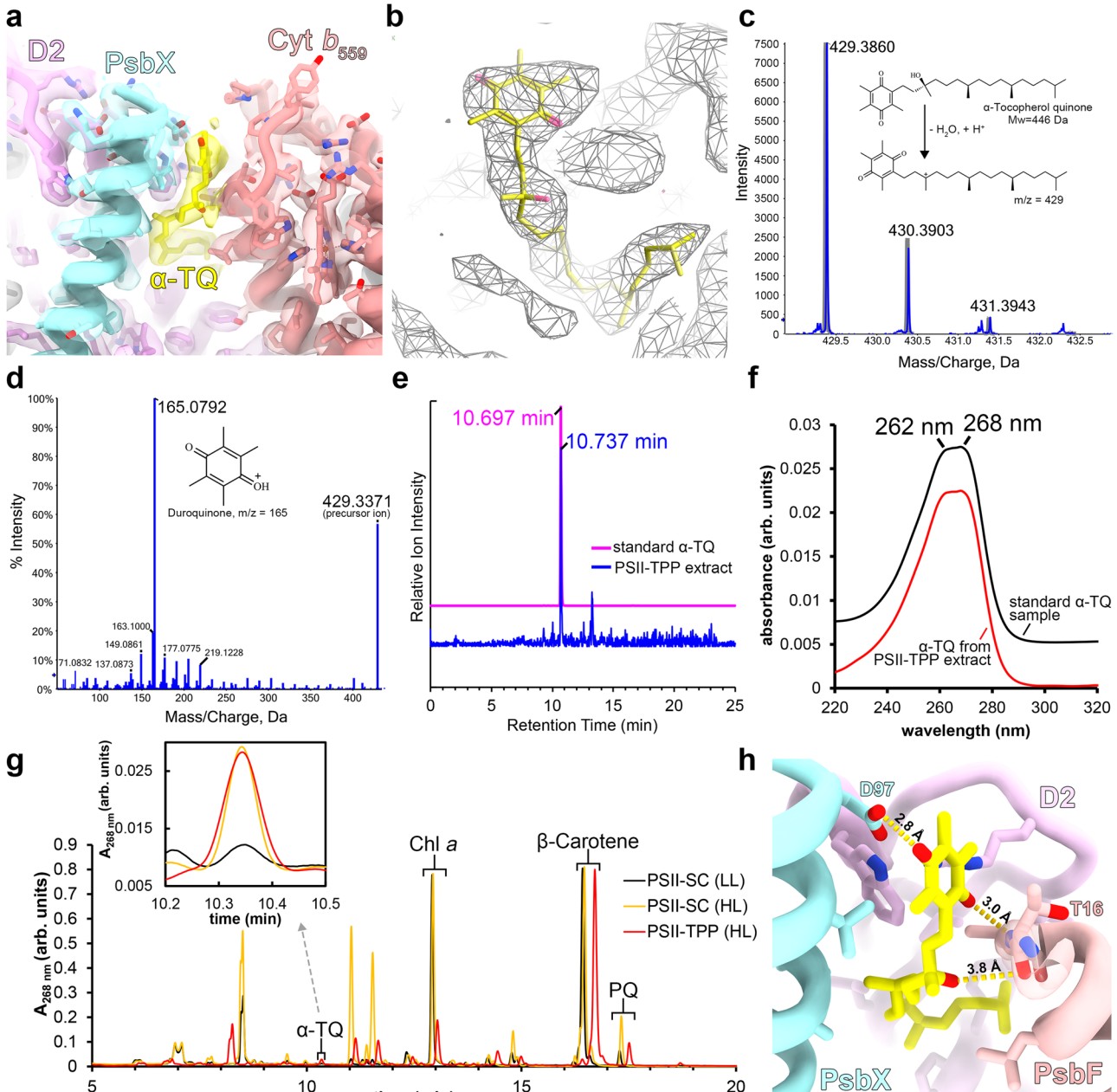

**Fig. 6 | An α-tocopherol quinone molecule located at an interstitial binding site between PsbX and PsbF. a** The local region of α-TQ-binding site in the structure of PSII-TPP complex superposed with the corresponding cryo-EM density. The α-TQ molecule is highlighted in yellow. **b** The model of α-TQ molecule fits well with the cryo-EM density. **c, d** Identification of α-TQ extracted with hexane from the PSII-TPP complex sample by using the liquid chromatography–mass spectrometry (LC-MS) method. The peaks in the primary-ion mass spectrometry (**c**) with m/z of 430.4 and 431.4 are isotope peaks of α-TQ. Characteristic precursor ion with a m/z of 429.4 and fragment ion with a m/z of 165.1 are detected in the secondary-ion mass spectrometry (MS$^2$) (**d**). In (**c**), the blue trace is the experimental isotope spectrum, while the gray one is the ideal isotope spectrum calculated from the chemical formula ($C_{29}H_{49}O_2$) as a reference. **e** Chromatograms of the hexane-extracted fraction from the PSII-TPP sample and the standard α-TQ sample. The relative precursor ion intensity is plotted against retention time. Due to the intricate composition of the extract, only the compounds with precursor ion m/z of 429 are shown. **f** The characteristic absorption spectrum of α-TQ from the high-light PSII-TPP sample and the standard α-TQ sample as detected by the photo-diode array (PDA) monitor. The absorption spectrum of the standard α-TQ sample was normalized and plotted as a black curve above the one from the PSII-TPP sample (red) for comparison. **g** The high-performance liquid chromatography (HPLC) profiles for the organic ligands extracted by n-hexane from three different PSII complexes. The HPLC profiles of the extracts from the PSII-LHCII supercomplex samples of the low-light and high-light-adapted cells, namely PSII-SC (LL) and PSII-SC (HL), are included for comparison with that of the extract from the PSII-TPP (HL) sample. The chromatographic traces were normalized based on the absorption peak value of β-carotene. A zoom-in view of the absorption peaks corresponding to α-TQ is shown on the upper left side. **h** Interactions between α-TQ and the adjacent protein subunits. Source data are provided as a Source Data file.

protonatable aspartic acid residue may serve as a proton donor for the quinone group of α-TQ during the reduction of α-TQ. The redox potentials of Cyt $b_{559}$ in PSII ranges from −150 to +390 mV depending on the conditions, and conversion of the high-potential Cyt $b_{559}$ to the low-potential form occurs under high-light and other conditions[26]. The

low-potential form of Cyt $b_{559}$ can receive electrons from one of the pheophytin molecules and reduce oxygen into the hazardous superoxide ($O_2^{\cdot-}$)[49–52]. Previous studies have shown that α-TQ is capable of oxidizing the low-potential Cyt $b_{559}$ (ref. 53), and the product may be α-tocopherol quinol (α-TQH$_2$)[54]. Thereby, the generation of superoxide

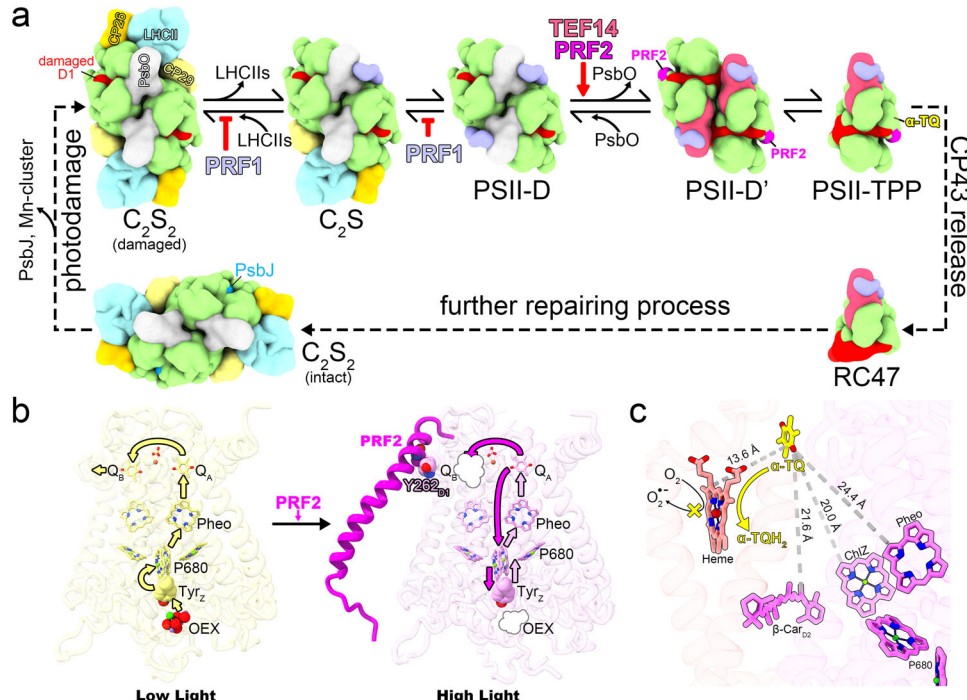

**Fig. 7 | A hypothetical working model on the functional roles of PRF1, TEF14, PRF2 and α-tocopherol quinone in the PSII repair process. a** Under the high-light conditions, the dissociation and association/reassociation of LHCIIs with PSII core are in equilibrium. PRF1 promotes the disassembly of the PSII supercomplex by binding to the damaged PSII core and blocks the re-association of CP29. Thereby, the equilibrium is driven towards the dissociation direction. TEF14 promotes the dissociation of PsbO by competing with part of its binding site on the lumenal surface. When PsbO is dissociated from the PSII core, part of the D1 lumenal loop regions, which are candidates for proteolytic degradations, are exposed to the bulk solution. Further release of the CP43 module through unknown mechanisms may fully expose the damaged D1 protein to the proteases, ensuring the degradation and removal of the damaged D1 subunit. The view is from lumenal side and along membrane normal. **b** PRF2 may induce blockage of the $Q_B$-binding site by interacting with Tyr262 of D1 subunit and causing conformational changes on the Q261-F265 loop of D1. For clarity, the tail groups of plastoquinone (PQ), pheophytin and chlorophyll *a* are omitted, and only the D1, D2 and PRF2 subunits are shown. The view is along membrane plane approximately. Under high light, when the Mn cluster becomes damaged and dissociates from the reaction center, the blockage of $Q_B$ site may promote direct charge recombination between $Q_A^-$ and $Tyr_Z^+$, preventing the formation of long-lived P680$^+$. Pheo: pheophytin; OEX: the $Mn_4O_5Ca$ cluster of the oxygen-evolving complex in PSII. **c** Minimal distances between the conjugate π-system of α-TQ head group and the adjacent electron transporting cofactors. A short distance of 13.6 Å between α-TQ and the porphyrin ring of the Cyt $b_{559}$ heme is favorable for electron tunneling between them. A putative working model for α-TQ to oxidize heme and prevent the generation of superoxide is shown as the yellow and dark arrows with different widths.

can be prevented. Taken together, our findings suggest that there might be a photoprotective electron transport pathway involving α-TQ and Cyt $b_{559}$ during the PSII repair process (Fig. 7c).

## Discussion

In this study, we combined structural, biochemical, and genetic approaches to reveal the functions of three protein factors and a small organic molecule in the PSII repair process, namely TEF14, PRF1, PRF2, and α-TQ. Among the three proteins, TEF14 is similar to *At*MPH2 in both sequence and function. Our results reveal the binding site of TEF14 on an intermediate complex of the PSII-repair process and further demonstrate its functions in facilitating degradation of damaged D1 protein and dissociation of PsbO from the damaged PSII core complex. Besides, PRF1 and PRF2 are single-pass transmembrane proteins discovered and characterized through biochemical, structural and functional analyses in this study.

The PRF1 protein from *C. reinhardtii* has homologs in cyanobacteria, such as the previously reported *Thermosynechococcus elongatus* Psb34 (ref. [27], named as Psb36 in the other report[55], also known as Tsl0063, Supplementary Fig. 9a). PRF1 shares 30.2% sequence identity and 54.7% similarity (excluding transit peptide sequence) with *Te*Psb34 (Supplementary Fig. 9a). Besides, the sequence and structural feature of PRF1 are also similar to those of the land-plant specific MPH1[56] (Maintenance of PSII under High light 1) (Supplementary Fig. 9a, b). The previous reports indicated that the cyanobacterial Psb34 binds to an intermediate complex during the PSII assembly

process[27,55], and may function to substitute the high-light-inducible proteins (Hlips, HliA/B)[57]. Hlip proteins in cyanobacteria are the ancestors of the light-harvesting proteins (LHCP) in the green lineage, including Lhcb4 (CP29)[58]. In *A. thaliana*, MPH1 was proposed to be involved in the stabilization and/or protection of various PSII complexes (PSII supercomplex, PSII dimer and monomer) from photodamage[56]. As for PRF1, its function may resemble that of Psb34 by associating with CP47 and preventing the peripheral light-harvesting complexes (such as CP29/Hlip) from associating with the PSII core.

The PSII repair cycle is an intricate multi-step process fine-controlled by more than 60 protein factors[13]. These factors may bind to the intermediate complexes following a certain chronological order so that the process can proceed in an orderly manner. Thus, it will be interesting to address when and how a specific factor binds and dissociates from the intermediate complex. In the structural model of the low-light adapted PSII-SC (PDB ID: 6KAC), the PsbJ protein occupies the cleft between Cyt $b_{559}$ and Ycf12 and blocks the entrance for PRF2 (Supplementary Fig. 10a). Putatively, the PsbJ subunit may be dissociated at an earlier step in the high light-adapted PSII complexes. The absence of PsbJ is usually related to the formation of an inhibited or intermediate form of PSII core[59–61]. Thus, we speculate that the release of PsbJ may be a signal for PSII to recruit PRF2 to the binding site so that it could induce the formation of a photoprotective state for PSII core. A recent work[27] suggested that Psb28 functions at the late stage of the PSII repair cycle when CP43 is assembled to a newly synthesized

RC47 complex. The space occupied by Psb28 overlaps largely with the binding site for the carboxy-terminal region of PRF2 (Supplementary Fig. 10b), indicating that the two factors may not bind simultaneously on the PSII complex. It is possible that the release of PRF2 is stimulated by the binding of Psb28, or occurred at a step before the binding of Psb28. It is noteworthy that PRF2 and Psb28 share a similar function in terms of deforming the $Q_B$ binding site. Thus, the substitution of PRF2 by Psb28 would help to maintain the PSII core complex in the inactive and photoprotective state from the early to late stages throughout the PSII repair process.

To account for the mechanism underlying the functions of TEF14, PRF1 and PRF2, we propose a hypothetical working model (Fig. 7) by considering the structural and functional analysis results described above. Under high-light conditions, the peripheral antennae dissociate from the PSII-core, and PRF1 may inhibit their re-assembly by occupying the exposed CP29 binding site (Fig. 4a, Supplementary Fig. 6a, b). TEF14 binds to the lumenal surface of CP47, likely at a stage when the peripheral antennae have dissociated from the PSII-D (Figs. 3c and 4c). The association of TEF14 with CP47 may facilitate dissociation of PsbO (Fig. 3b, d, g), destabilize the PSII dimerization interface and promote the disassembly of PSII-D into PSII-M (Fig. 3e). Further disassembly of PSII-M into RC47 would facilitate proteolytic degradations of D1 protein[62]. The association of PRF2 with PSII-M may restrict the accessibility of the $Q_B$ site to PQ, so that the photoprotective charge recombination between $Q_A^-$ and $Tyr_Z^+/P680^+$ may occur when the donor (Mn cluster) and the acceptor ($Q_B$ site) sides are both inactivated[27] (Fig. 7b). The formation of $P680^+$ cation radicals will likely be reduced and further damages of the PSII core subunits can be prevented as a result. The α-TQ molecule may oxidize the adjacent Cyt $b_{559}$ in competition with molecular oxygen, thus reducing the generation of toxic superoxide anion at the Cyt $b_{559}$ site (Fig. 7c).

## Methods

### Strains and growth conditions

*Chlamydomonas reinhardtii* control strain CC-5325 (WT) was obtained from the Chlamydomonas Resource Center (https://www.chlamycollection.org/). The *pph1;pbcp* double mutant strain was a kind gift from M. Goldschmidt-Clermont at the University of Geneva[35]. The *tef14*, *prf1*, and *prf2* gene knockout strains were generated by the CRISPR-Cas9 method. All strains were initially grown in Tris-acetate-phosphate (TAP) medium[63] under dim light (20 μmol photons $m^{-2} s^{-1}$) at 23 °C until they reached the mid-log growth phase, and subsequently exchanged into high-salt minimal (HSM) medium[64] for high light treatment unless otherwise stated. For the preparation of PSII-M-TEF14-PRF1-PRF2 (PSII-TPP) sample, CC-5325 and *pph1;pbcp* strains were treated under high light intensity of 330 μmol photons $m^{-2} s^{-1}$ for 24 h. For biochemical characterization of the *prf1* strain, high light durations were shortened to 4 h and the high light intensity was set to 600 μmol photons $m^{-2} s^{-1}$. The high light conditions for the rest strains remained to be 330 μmol photons $m^{-2} s^{-1}$ for 24 h. High light conditions for control strain (CC-5325) were kept in line with the mutant strains. For PSII quantum efficiency characterization, CC-5325, *tef14*, *prf1*, and *prf2* strains were treated under high light with an intensity of 750 μmol photons $m^{-2} s^{-1}$ for 8 h.

### Generation of knock-out *Chlamydomonas* strains

The *prf1, prf2,* and *tef14* mutants were generated by CRISPR/RNP-based gene editing method following a previous protocol with slight modifications[65]. In general, in vitro assembled Cas9-guide RNA (gRNA) ribonucleoproteins (RNPs), together with exogenous double-stranded donor DNA carrying antibiotic-resistance expression cassette, were delivered into the *C. reinhardtii* cells by electroporation. The mutants were screened and confirmed by PCR and western blot analysis (Supplementary Fig. 11). Two gRNA target sites were tested for each gene. gRNAs were designed using CRISPR-P v2.0 guide RNA tool[66] (http://

crispr.hzau.edu.cn/CRISPR2/) (Supplementary Table 2), and were synthesized by GenScript (SafeEdit sgRNA). 1 μL sgRNA (40 μM) was incubated with 2 μL of homemade *Streptococcus pyogenes* Cas9 (*Sp*Cas9) (10 mg/mL) in NEBuffer™ 3.1 (New England Biolabs, B7203) at 37 °C for 15 min in a final volume of 10 μL. The pre-assembled active RNP complex was further used for transformation. The paromomycin-resistance donor DNAs for *prf1* and *tef14* were amplified from the pMJ016c plasmid[67]; the hygromycin-resistance donor DNA for *prf2* was amplified from the pRAM118 plasmid[68]. Each donor DNA contained an integral antibiotic-resistance cassette[69,70] and gene-specific homology arms of 50 nt. Primers used for donor DNA amplification are listed in Supplementary Table 3. The *C. reinhardtii* strain CC-5325 (*cw15*, mt-) was used in this study. Cells were grown in 250 mL Tris-acetate-phosphate (TAP) medium at 25 °C under continuous cool-white fluorescent light (~100 μmol photons $m^{-2} s^{-1}$) with 120 rpm shaking, until they reached the mid-log phase ($OD_{750}$ around 0.5−0.6). For transformation, cells were incubated at 40 °C for 30 min with gentle agitation. Cells were then collected by centrifugation at $1,820 \times g$ for 1 min, washed once with 10 mL TAP + 2% sucrose, and finally resuspended in the same buffer at a concentration of $2.0 \times 10^8$ cells/mL. 110 μL of concentrated cells were then mixed with 1 μg of donor DNA and 10 μL of Cas9/gRNA RNP to give a final volume of 125 μL. The mixture was delivered into the cells by electroporation in a Bio-Rad 0.2 cm-gap cuvette using a BTX ECM 600 at 350 V, 25 Ω, and 600 μF. Immediately after electroporation, cells were incubated at 16 °C for 1 h and then transferred to 8 mL TAP + 2% sucrose in a 15 mL tube and incubated at room temperature in dark for 24 h. Following recovery, cells were collected by centrifugation at $1820 \times g$ for 1 min, resuspended with 2 mL TAP + 0.4% low melting point (LMP) agarose (<37 °C), and pipetted onto TAP + 1.5% agar containing either 20 μg/mL paromomycin or 30 μg/mL hygromycin B. The plates were placed under low light (~30 μmol photons $m^{-2} s^{-1}$) at 25 °C for about 7−10 days.

### Genotyping of potential mutants

Antibiotic-resistant transformants were arrayed on agar plates and were grown under continuous white light for 4 days. A small amount of each cell line was mixed with 20 μL of 10 mM EDTA and then heated to 99 °C for 10 min to generate crude genomic DNA extraction for screening of cell lines via PCR at target site. Primers used for genotyping are listed in Supplementary Table 4. The successful mutations of *TEF14*, *PRF1*, and *PRF2* were verified at multiple levels as shown in Supplementary Fig. 1.

### Isolation and characterization of PSII complexes from WT and *pph1;pbcp* strains

Thylakoid membranes were isolated according to a protocol reported previously[71]. In detail, cells treated under low light or high light conditions were collected and resuspended in the lysis buffer (25 mM 4-(2-hydroxyethyl)-1-piperazineethanesulfonic acid (HEPES), 0.3 M sucrose and 5 mM $MgCl_2$), and passed five times through a high-pressure homogenizer device (AH-2010, ATS Engineering) at a pressure of 400 bar. The homogenate was centrifuged, and the thylakoid-containing pellet was washed once with the washing buffer (5 mM HEPES, 0.3 M sucrose and 10 mM EDTA). The thylakoid membranes were further separated from contaminants through a discontinuous sucrose density gradient (SDG) with three layers containing 0.5, 1.3, and 1.8 M sucrose, respectively. For high-light treated cells, the thylakoid membrane sample was solubilized with 1.0% (w/v) α-DDM at 4 °C for 10 min at a chlorophyll concentration of 0.5 mg ml$^{-1}$. For low light-adapted cells, the PSII-M fraction solubilized by α-DDM can barely be detected, so we used 0.4% β-DDM instead of 1.0% α-DDM for better yield of the PSII-M fraction. The low-light PSII-M samples are only used as controls representing the "background" protein composition of PSII-core during SDS-PAGE analysis. Insoluble fractions were discarded after centrifugation at $10,000 \times g$ for 5 min. Photosynthetic complexes

were separated by SDG ultracentrifugation at $256{,}000 \times g$ for 12 h at 4 °C, with the matrix containing 5–35% sucrose, 25 mM 2-(N-morpholino) ethanesulfonic acid (MES) pH 6.5 and 0.02% α-DDM (or β-DDM). Bands containing PSII-M were collected for further study.

## Characterization of PSII-TPP complex

The SDG band containing PSII-M was collected and the protein composition of the fraction was analyzed by 13% sodium dodecyl sulfate-polyacrylamide gel electrophoresis (SDS-PAGE) containing 6 M urea. The gels were stained with Coomassie Brilliant Blue R250 or a mass spectrometry compatible silver staining kit (Protein Stains K, Sangon Biotech). The identities of proteins were further verified by mass spectroscopy analysis of the in-gel protease-digestion products of the excised SDS-PAGE bands. For estimating the relative abundance of TEF14, PRF1 and PRF2/PRF2′ vs the major subunits of PSII core, the SDS-PAGE gel was stained with the Sypro Ruby dye which exhibits high sensitivity and a broad linear dynamic range (1–2 ng to 1–2 µg) for quantification of proteins on the SDS-PAGE gel[72]. The gel was scanned using the Typhoon FLA 9500 imager at an excitation wavelength of 473 nm, and the fluorescence was detected using a filter with passing wavelength greater than 665 nm (long-pass red, LPR filter). The fluorescence intensity was calculated using the ImageJ software[73]. The relative amount (in arbitrary unit) of protein in the designated band is quantified as the fluorescence intensity of the band divided by the corresponding protein molecular weight. The molar ratios were calculated as the ratios between the relative amounts of the target protein (TEF14, PRF1 or PRF2) and that of CP47.

## Protein mass spectrometry analysis

The bands on the SDS-PAGE gel were cut and treated using dithiothreitol and iodoacetamide, followed by proteolytic digestion with trypsin or chymotrypsin overnight. The resulting peptide fragments were desalted by using the C18 ZipTip pipettes (Millipore) and then mixed with a matrix of α-cyano-4-hydroxycinnamic acid for spotting on the plate. Finally, the analysis was carried out by using the matrix-assisted laser desorption/ionization time-of-flight mass spectrometry (MALDI-TOF/TOF Ultraflextreme, Brucker, Germany) controlled by FlexControl v.3.4 software package. The instrument was externally calibrated by using the Bruker peptides calibration kit. The mass spectra were scanned in the range of m/z 700–5000. The spectra were recorded in the reflection positive ion mode (laser intensity 95%, ion source 1 = 20.00 kV, ion source 2 = 17.75 kV, lens = 7.6 kV, detector voltage = 2127 V, and pulsed ion extraction = 180 ns). Each spectrum corresponded to an ion accumulation of 500–1000 laser shots randomly distributed on the spot. For the MALDI-TOF/TOF-MS analysis, precursors were accelerated and selected in a time ion gate, after which fragments arising from metastable decay were further accelerated in the LIFT cell and detected after passing the ion reflector. The database searches were conducted by using the Peptide Mass Fingerprinting or the MS/MS Ion Search pages on the local Mascot website. Identification of proteins was based on the primary or secondary mass spectra of the peptide fragments generated after digestion. The search parameters included the trypsin/chymotrypsin digestion with up to two missed cleavage sites, cysteine alkylation as a fixed modification, and methionine oxidation as a variable modification. The Uniprot proteome database of *Chlamydomonas reinhardtii* was used for the identification process. Two biological replicates were performed for the mass spectrometry analysis of the TEF14, PRF1, PRF2, and PRF2′ proteins.

## Heterologous expression and purification of TEF14 and the carboxy-terminal fragment of PRF1 (PRF1-C)

The cDNA sequence of TEF14 following a Strep-His₆-Strep tag at the amino-terminal was codon-optimized and synthesized (GenScript). The synthesized gene was further cloned into pET-15b vector between the restriction sites of *Nco*I and *Blp*I. The chloroplast and thylakoid lumen transit peptides were predicted by TargetP-2.0[74] and removed from the sequence. The carboxy-terminal fragment of PRF1 (Thr165-Leu189) was codon optimized and cloned into pMAL-c4X vector between the restriction sites of *Sac*I and *Hind*III. *E. coli* strain C43(DE3) was used as the host strain for heterologous expression of the target proteins. The transformants were cultured in the Luria-Bertani (LB) medium supplemented with 100 µg/mL ampicillin to an $OD_{600\,nm}$ of 0.8–1.0 at 37 °C. Subsequently, IPTG was added to the culture at a final concentration of 0.5 mM to induce protein expression for 2 h. The harvested cells were suspended in the lysis buffer (30 mM Tris-HCl, pH 7.5, 300 mM NaCl, 5% Glycerol) and disrupted by a high-pressure homogenizer (ATS Engineering Inc) at 900 bar for 4 cycles. For the purification of TEF14, the TALON metal affinity resin was used and the target protein was eluted using buffer containing 30 mM Tris-HCl, pH 7.5, 300 mM NaCl, 5% glycerol and 90 mM imidazole. For purification of the MBP-tagged PRF1-C, the Dextrin Sepharose High Performance resin (GE Healthcare) was used and the elution buffer contains 25 mM Tris-HCl, pH 7.5, 200 mM NaCl, 1 mM EDTA and 10 mM maltose.

For the in vitro assay on the function of TEF14 (Fig. 3f, g), the purified recombinant TEF14 protein was added to the solubilized thylakoid (0.5 mg/mL Chl) at a final concentration 25 µg/mL and then incubated for 10 min at 4 °C. The mixture was subjected to sucrose density gradient ultracentrifugation in a medium containing 5–30% sucrose, 25 mM 2-(N-morpholino) ethanesulfonic acid (MES) pH 6.0, 75 mM NaCl and 0.02% α-DDM. The ultracentrifugation was carried out at $256{,}000 \times g$ for 12 h in a SW 40 Ti Swinging-Bucket Rotor loaded in a Coulter Optima XPN-100 centrifuge (Beckman). After ultracentrifugation, the fractions corresponding to PSII-M and PSII-D were extracted and analyzed through the western blots of the SDS-PAGE gels probed by anti-PsbO (PHY0094A, PhytoAB) and anti-CP47 (As04038, Agrisera) antibodies respectively. The western blots of PsbO and CP47 from the same SDS-PAGE gel are imaged in two separate batches (one batch for PsbO and the other for CP47) by using the ChemiScope 3500 Mini Imaging System (Clinx Science Instruments). The bands on the western blot images were analyzed and quantified by using the ImageJ software[73].

## Isothermal titration calorimetry (ITC) assay

The TEF14 and PRF1-C protein samples eluted from the affinity chromatography were further purified and buffer-exchanged through the gel-filtration chromatography using a Superdex 200 Increase 10/300 GL column (GE Healthcare). The gel filtration buffer contains 30 mM Tris-HCl, pH 8.3, 300 mM NaCl and 5% Glycerol. For ITC assay, 40 µL of TEF14 with a concentration of 330 µM was titrated to 250 µL of 33 µM PRF1-C on a MicroCal iTC200 device (Malvern Panalytical). The titration was performed at 16 °C with 20 injections. Void buffer was titrated with the TEF14 sample as a reference for the subtraction of dilution heat. Data analysis was performed with the MicroCalITC200 software.

## Western blot analysis

The SDS-PAGE gels were transblotted onto PVDF membranes using a Trans-Blot Turbo Transfer System (Bio-Rad). Antibodies against the carboxy-terminal fragment of D1 (AS05084) and the CP47 apoprotein (AS04038) were purchased from Agrisera. Antibodies against the amino-terminal of D1 (PHY0057) and the PsbO subunit (PHY0094A) were purchased from PhytoAB. Multi-clonal antibody against the TEF14 subunit was prepared by ABclonal (Wuhan, China) using the purified recombinant *Cr*TEF14 protein expressed in *E. coli*. The amino-terminal Strep-His₆-Strep tag was excised before the protein was used for immunization of rabbits.

## Measurement of maximal quantum efficiency of PSII

The *C. reinhardtii* cells were centrifuged and resuspended in fresh HSM medium to a chlorophyll concentration of 15 µg/mL. All samples were dark-adapted for 10 min prior to the fluorescence measurements. The

potential maximum of PSII quantum yield ($F_v/F_m$) was determined by using a pulse amplitude-modulated fluorimeter (Dual-PAM 100, Walz). Saturating light (6000 µmol photons·m$^{-2}$·s$^{-1}$, 250 ms) and red measuring light were used throughout all measurements. The $F_v/F_m$ values were calculated according to the formula $F_v/F_m = (F_m − F_O)/F_m$. $F_m$, and $F_O$ represents the maximum and minimum fluorescence respectively[75].

### Cryo-EM grid preparation and data acquisition

Sucrose was removed from the PSII-TPP sample prior to grid preparation. The sample collected from SDG was diluted ten times with a sucrose-free buffer (25 mM MES pH 6.5 and 0.02% α-DDM) and then concentrated 10–20 times in a 100-kDa molecular-weight cut-off centrifugal filter unit (Amicon Ultra-15, Merck Millipore). The dilution and concentration process were repeated three times. Subsequently, 3 µL of the concentrated sample (with a concentration of 2.5 mg ml$^{-1}$ Chl) was loaded onto a H$_2$/O$_2$ glow-discharged holey carbon grid (Quantifoil 300-mesh, R1.2/1.3). The grid was blotted for 3 s with a force level of 2 at 22 °C and humidity of 100%, flash-frozen in liquid ethane and transferred to liquid nitrogen for storage. Micrographs were collected using the SerialEM software suite[76] on a 300-kV Titan Krios electron microscope (Thermo Fisher Scientific) equipped with a Gatan K2 Summit direct electron detector and a Gatan GIF Quantum energy filter (20 eV). A total of 5189 micrographs were collected at a defocus range of −1 to −1.5 µm at ×130,000 magnification and a pixel size of 0.52 Å in the super-resolution mode. Images were recorded by beam-image shift data collection methods[77]. Micrographs were exposed at a total dose of 60 e$^-$/Å$^2$ and dose fractionated into 40 frames.

For the preparation of PSII-M from the *prf2* mutant, 1 liter of the mutant cells was treated under high light at an intensity of 330 µmol photons m$^{-2}$ s$^{-1}$ for 24 h. The protocols for the thylakoid preparation, solubilization and complex separation were the same as those used in preparing the PSII-TPP complex from the *pph1;pbcp* strain. After the sucrose density gradient ultracentrifugation, the green band corresponding to the PSII-M migrating between the bands of LHCII trimer and the PSI-LHCI supercomplex was collected. After removing the sucrose from the sample and concentrating it to 2.2 mg ml$^{-1}$ Chl, 3 µL of the sample was loaded onto a H$_2$/O$_2$ glow-discharged holey carbon grid (Quantifoil 300-mesh, R1.2/1.3). The grid was blotted for 4 s with a force level of 2 at 22 °C and humidity of 100%, flash-frozen in liquid ethane and transferred to liquid nitrogen for storage. Micrographs were collected using the SerialEM software on a 300-kV Titan Krios electron microscope (Thermo Fisher Scientific) equipped with a Gatan K3 direct electron detector without energy filter. The beam-image shift method was applied during the data collection process. A total of 4321 micrographs (with a total dose of 60 e$^-$/Å$^2$ fractionated into 40 frames for each micrograph) were collected at a defocus range of −1 to −1.5 µm at ×22,500 magnification and a pixel size of 0.535 Å in the super-resolution mode.

### Cryo-EM data processing, classification, and reconstruction

The 40 frames in each movie stack were aligned, dose weighted and summed using MotionCor2[78] and binned to a pixel size of 1.04 Å. The contrast transfer function (CTF) parameters of the motion-corrected micrographs were estimated using CTFFIND4.1[79]. Further image-processing steps were performed in RELION v.4.0[80], including reference-based particle autopicking, two-/three-dimensional classifications, three-dimensional auto refinement, CTF refinement and Bayesian particle polishing. The detailed data-processing workflow is shown in Supplementary Fig. 2. To overcome the low occupancy of TEF14, PRF1, and PRF2 in the PSII-M complex, focused classifications were employed without alignment using local masks of TEF14 and PRF2, respectively. The occupancy of PRF1 density is highly correlated with that of TEF14, and does not need a separate classification process. Local maps were generated in UCSF Chimera v.1.14[81], and were further processed into masks using RELION. The selected particles with the

best PSII-core, TEF14/PRF1 and PRF2 densities were focused refined and sharpened, and the resulting local maps were further combined using the Combine Focused Maps program in Phenix v.1.19 (ref. 82). Map sharpening was done in Local MonoRes[83] and LocalDeblur[84] through the Scipion v.3.0[85] pipeline. Local resolutions were estimated by RELION v.4.0. The local resolution of the final composite map ranges from 2.5–7.0 Å (Supplementary Fig. 2e, f). The orientational distribution of the particles was assessed by CryoEF[86].

For the cryo-EM data of the PSII-M from the *prf2* mutant, an initial data set with 1,586,163 particles were auto-picked by using the RELION v4.0 program after motion correction and CTF estimation. After one round of 2D classification followed by two rounds of 3D classification, 228,397 particles with clear PSII-M features were selected, and further refined to obtain the map of PSII-M-PRF2′ complex at an overall resolution of 2.9 Å. To improve the density of PRF2′, we employed one round of focused 3D classification without alignment by using a local mask around the PRF2′. A dataset of 159,928 particles with improved PRF2′ density was selected, and was refined to an overall resolution of 3.1 Å (Supplementary Fig. 12a, b). The orientational distribution of the particles is shown in Supplementary Fig. 12c. The local resolution of the final composite map ranges from 2.9–4.2 Å as estimated by RELION 4.0 (Supplementary Fig. 12d). The model for PRF2′ was built in Coot v.0.9.8 manually, and the overall model was further refined in Phenix v.1.19 using the method same as that used for the PSII-TPP complex (Supplementary Fig. 12e, f).

### Model building and refinement

The model for PSII core monomer region of a previously published *C. reinhardtii* PSII-SC model (PDB ID: 6KAC) was extracted and used as initial model. The model was docked into the final composite map with UCSF Chimera, and manually adjusted in Coot v.0.9.8[87]. The cryo-EM densities of the Mn-cluster, PsbO, PsbP, PsbQ, PsbJ, PsbR, the unknown stromal protein and the predicted lumenal protein are absent in the PSII-TPP complex map, so the corresponding models were removed. The initial atomic models for TEF14 and PRF1 were generated by trRosetta[88] predictions and adapted from the atomic model of *Te*Psb34 (PDB ID: 7NHP) with its sequence mutated to that of *Cr*PRF1, respectively. The initial models were docked into the density map in UCSF Chimera and manually adjusted in Coot to fit with the cryo-EM density. The structural models for PRF2 and the carboxy-terminal region of PRF1 located in the thylakoid lumenal side were built manually in Coot based on the density features. The manually adjusted model was further subjected to real-space refinement in Phenix v.1.19, and the geometric restraints for the co-factors and chlorophyll-ligand coordination bond parameters were supplied during refinement. The automatic real-space refinement and manual inspection processes were carried out iteratively until the geometries of the final structures converged to a reasonable range, as assessed by MolProbity[89]. A summary of the statistics for data collection and structure refinement is provided in Supplementary Table 1. High-resolution images for publication were prepared by using ChimeraX v.1.5 (ref. 90). The structural model of *At*MPH1 shown in Supplementary Fig. 9b was predicted by AlphaFold2 (ref. 91).

### Extraction and identification of α-tocopherol quinone

n-Hexane was used to extract prenyllipids and other highly hydrophobic compounds from the peripheral regions of the PSII-TPP sample[92,93]. Approximately 100 µL of PSII sample with a chlorophyll concentration of 1.5 mg/mL was mixed and vortexed with an equal volume of n-hexane. The upper hexane phase after extraction was light yellowish green, indicating that most of the chlorophylls associated with the internal regions of PSII complex remained unextracted. The hexane phase was carefully collected and dried under nitrogen gas flow, and the remaining yellowish-green precipitate was further dissolved by a mixture of chloroform and methanol (2:1, v/v). The sample

was analyzed by using a high-performance liquid chromatograph (HPLC) (Thermo Scientific Dionex Ultimate 3000) coupled to quadrupole time-of-flight tandem mass spectrometer (SCIEX TripleTOF 5600). The HPLC separation was carried out on an ACQUITY UPLC CSH C18 reversed-phase column (2.1 mm × 100 mm, 1.7 mm, Waters) at a flow rate of 0.2 mL/min. The temperature of the column was maintained at 40.0 °C. The mobile phase consisted of two components, namely phase A (methanol/acetonitrile/15 mM ammonium acetate aqueous (1:1:1, v/v/v)) and phase B (80% (v/v) 2-propanol and 20% (v/v) methanol containing 5 mM ammonium acetate). The column was eluted for 0-1.0 min with 10% B, 1.1–6.0 min with a linear gradient from 10% to 60% B, 6.1-18.0 min from 60% to 100% B, and 18.1–20.0 min held at 100% B, followed by 20.1–25 min from 100% to 10% B. A standard sample of α-TQ compound (CDS000038, Merck) was also applied in the liquid chromatography separately as a reference (Fig. 6e). The MS analysis was performed in positive ion mode (electrospray ionization/ESI) with a scan range of 100−1250 m/z for MS, or 50-1250 m/z for MS/MS. Ion spray voltage was set at 5.5 kV in the positive mode, and the source temperature was 600 °C. The ion source gases 1 and 2 were both set at 60 psi, and the curtain gas was set at 35 psi. In the product ion mode, the collision energy was set at 35 V. Analysis of the LC-MS data was performed using Peakview (v.2.1), and visualized using Matplotlib. The identification of α-tocopherol quinone (α-TQ) was achieved on the bases of three criteria, namely precursor ion m/z, fragment ion m/z, and isotope spectrum. The precursor ion of α-TQ appears at m/z of 429 rather than 447 due to the dominant loss of water effect under positive ion mode[47] (Fig. 6c). The fragment ion with aa m/z of 165 matches well with the duroquinone head group of α-TQ according to the previous report[47] (Fig. 6d). The experimental isotope spectrum (blue trace in Fig. 6c) also matches well with the ideal one (gray traces).

For the absorption spectra and HPLC data shown in Fig. 6f, g, the samples containing α-tocopherol quinone were extracted from the PSII complexes purified from high-light or low-light adapted cells by using n-hexane. The different components of the samples were separated on a Waters ACQUITY UPLC BEH C18 column (1.7 μm, 100 × 2.1 mm) connected to the HPLC system (Waters ACQUITY UPLC H-class PLUS with photodiode array detector (PDA) & QDa detectors) and eluted at a flow rate of 0.2 mL/min. The temperature of the column was maintained at 40.0 °C. The mobile phase consisted of two components, namely phase A (methanol/acetonitrile/ 15 mM ammonium acetate aqueous (1:1:1, v/v/v)) and phase B (80% (v/v) 2-propanol and 20% (v/v) methanol containing 5 mM ammonium acetate). The sample loaded on the column was eluted for 0−1.0 min with 10% B, 1.1–6.0 min with a linear gradient from 10% to 60% B, 6.1–18.0 min from 60% to 100% B, and 18.1–20 min held at 100% B, followed by 20.1–25 min from 100% to 10% B. A standard sample of α-TQ compound (CDS000038, Merck) was also applied in the liquid chromatography separately as a reference. The absorption spectra of each individual peaks were collected from the PDA equipped in the liquid chromatography system. To estimate the stoichiometry of α-TQ per PSII-TPP complex, β-carotene is chosen as a relatively specific internal reference for estimating the amount of PSII as both α-TQ and β-carotene are highly soluble in n-hexane[94,95] and accessible to the extracting reagents. The maximum absorption wavelength of α-TQ (268 nm) was used to monitor the compounds eluted from the HPLC column. The molar extinction coefficient ($\varepsilon$) of α-TQ and β-carotene are 18.2 mM$^{-1}$ cm$^{-1}$ (ref. 96) and 20.8 mM$^{-1}$ cm$^{-1}$ (based on the data reported by ref. 97), respectively. The molar amount (in arbitrary unit) of α-TQ and β-carotene in the sample was estimated by dividing the absorption peak area by the corresponding $\varepsilon$ value (according to the Beer's law expressed as $c = A/\varepsilon L$; $c$, concentration; $A$, absorbance; $\varepsilon$, molar extinction coefficient; $L$, light path length in centimeters). Considering that there are ten β-carotene molecules associated with one PSII-TPP complex, the molar ratio of α-TQ $vs$ PSII-TPP is estimated as the molar amount

of α-TQ divided by one-tenth of the molar amount of β-carotene. The abundance of α-TQ per PSII-core monomer was estimated to be 0.23, 0.28, and 0.05 for the PSII-TPP complex, high-light adapted PSII-SC and low-light adapted PSII-SC, respectively.

## Quantitative reverse transcription polymerase chain reaction (RT-qPCR)

WT *C. reinhardtii* cells (strain CC-5325) were initially cultured in TP medium in triplicate under low-light (30 μmol photons m$^{-2}$ s$^{-1}$) conditions until the cultures reached log phase. Subsequently, they were moved to a growth chamber with high light (300 μmol photons m$^{-2}$ s$^{-1}$) illumination for 12 h. Sampling was conducted every two hours over the course of high-light exposure to collect cells for RNA extraction. The total RNA was isolated from the *Chlamydomonas* culture samples (exposed to different durations of high light) by using QIAGEN RNeasy plant mini kit (QIAGEN, Valencia, CA) according to the manufacturer's instructions. The first-strand cDNA was generated via the reverse transcription reaction with the HiScript II 1st Strand cDNA Synthesis kit (Vazyme, China). Independent RT-qPCR reactions were performed using the same cDNA for *TEF14*, *PRF1* and *PRF2* genes using ChamQ Universal SYBR qPCR Master Mix (Vazyme, China). Gene-specific primers were designed for these genes using the Primer3 software[98] and were listed in Supplementary Table 5. The *CBLP* gene, encoding a G-protein β subunit-like polypeptide, was used for normalization as its transcript level is not affected by the high-light treatment. The polymerase chain reaction program consisted of an initial denaturation step at 95 °C for 5 min, followed by 40 cycles of denaturation at 95 °C for 10 s and annealing/ extension at 60 °C for 20 s. The relative transcription level values were presented as means ± SD from three biological replicates.

### Reporting summary
Further information on research design is available in the Nature Portfolio Reporting Summary linked to this article.

## Data availability
The previously published structural models of the *Chlamydomonas reinhardtii* PSII-LHCII C$_2$S$_2$ type supercomplex, the *Thermostichus vulcanus* PSII complex and the *Thermosynechococcus vestitus BP-1* PSII-I assembly intermediate complex can be accessed from the Protein Data Band under the accession codes 6KAC, 3WU2 and 7NHP, respectively. The composite cryo-EM map of the *Chlamydomonas reinhardtii* PSII-TPP complex and its corresponding atomic coordinates have been deposited in the Electron Microscopy Data Bank and the Protein Data Bank under the accession codes EMD-37133 and 8KDE respectively. The cryo-EM map of the *Chlamydomonas reinhardtii* PSII-PRF2′ complex and its corresponding atomic coordinates have been deposited in the Electron Microscopy Data Bank and the Protein Data Bank under the accession codes EMD-60026 and 8ZEE respectively. The mass spectrometry proteomics data generated in this study have been deposited to the ProteomeXchange Consortium via the PRIDE[99] partner repository with the dataset identifier PXD052618. All data analyzed during this study are included in this Article and its Supplementary Information. Source data are provided with this paper. The *tef14*, *prf1* and *prf2* mutant strains of *Chlamydomonas reinhardtii* generated in this work and other related materials/data are available from the corresponding authors upon request. Source data are provided with this paper.

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

## Acknowledgements

We thank Li-Hong Chen, Xiao-Jun Huang, Bo-Ling Zhu, Bing-Xuan Huang-Fu, De-Yin Fan, Long-Long Zhang and other staff members at the Center for Biological Imaging (CBI), Core Facilities for Protein Science at the Institute of Biophysics, Chinese Academy of Science (IBP, CAS) for the support in cryo-EM data collection; Zhen-Sheng Xie, Xiang Ding, Li-Li Niu and Mengmeng Zhang at the CAS Research Platform for Protein Sciences for the support in mass spectrometry data collection and analysis; Xiao-Bo Liang and Xiu-Ying Liu for technical support on biochemistry, cell culture, sample preparation; Yi-Xuan Liu for extraction of mRNA and preparation of cDNA sample from WT and mutant strains; Yuan-Yuan Chen for technical support on the isothermal titration calorimetry analysis. Xiao-Xing Huang for the assistance on measuring the UV absorbance of the sucrose density gradient fractions. We also thank Prof. Guo-Liang Xu for his Cas9 expression plasmid and Dr. Mei Li for discussion. This work is funded by the National Natural Science Foundation of China (31925024 to Z.L.), the Chinese Academy of Sciences Project for Young Scientists in Basic Research (YSBR-015 to Z.L.), National Natural Science Foundation of China (32170255 to X.L.), the Strategic Priority Research Program of CAS (XDB37020101 to Z.L. and XDB27020106), the National Key R&D Program of China (2017YFA0503702 to Z.L.), and the National Key R&D Program of China (No. 2019YFA0904604 to L.T.).

## Author contributions

A.L. purified the PSII-TPP complex, prepared the cryo-EM sample, solved the Cryo-EM structure and performed the biochemical analysis to study the function of *TEF14*, *PRF1* and *PRF2*. A.L. and Z.L. built and refined the atomic model, identified the TEF14, PRF1 and PRF2 protein, and analyzed the structure. T.Y. performed the CRISPR/RNP-based gene editing to generate the *tef14*, *prf1*, *prf2* mutants and carried out the HPLC and RT-qPCR experiments under X.L.'s supervision. X.P. and A.L. performed the maximal quantum efficiency ($F_v/F_m$) measurements under L.T.'s supervision. Y.W. expressed and purified the TEF14 and MBP-PRF1-C protein, and conducted the protein interaction analysis by isothermal titration calorimetry. A.L. designed the mass spectrometry study and prepared the sample for the MS analysis. A.L., T.Y., X.P., Y.W., and Z.L. wrote the initial draft. All authors are involved in discussing and editing the manuscript. Z.L. conceived and coordinated the research project.

## Competing interests

The authors declare no competing interests.
