## [Peer Review File · Nature Communications]

Reviewers' Comments:

Reviewer #1:

Remarks to the Author:

Li et al. revealed a cryo-EM structure of an intermediate complex (PSII-monomer-TEF14-PRF1-PRF2 complex) of the PSII repair cycle from *pph1;pbcp* *Chlamydomonas reinhardtii* mutant adapted to high-light stress. They also provided a biochemical and functional characterization of this complex, specifically of TEF14, PRF1, and PRF2. This characterization was supported by analysis of the generated knockout mutants and the purified recombinant TEF14 protein. In my opinion, the authors present a thorough study on the reparative processes of photosystem II. Thanks to the appropriately designed experimental approach, the presented description of the function of this complex is supported by the experiments performed. Thus, the described study will contribute significantly to our knowledge in the field of reparative processes of the photosynthetic apparatus, specifically photosystem II. In my opinion, the work is of great interest to a broad scientific community and the manuscript is suitable for a publication in *Nature Communications* after clarification of some points.

Major comments:

- In the Fig 1b, the presence of PRF2 in the *pph1;pbcp* mutant is hardly visible compared to WT. Can you comment on that with respect to the presence of this subunit in the cryo-EM structure? Further, there are additional unassigned bands in both WT and *pph1;pbcp* mutant below the TEF14 band. Do you have any idea what these bands represents?
- The proposed model (Fig. 7) considers a binding competition between TEF14, PRF1, and PRF2 factors with other PSII core subunits, especially under high-light conditions. Is there any information on how the levels of these factors are controlled, e.g. under optimal light conditions, to reduce binding competition and avoid a disassembly of PSII supercomplexes when it is not necessary?
- In the Results part (lines 239-250), a role of PRF1 in disassembly of PSII supercomplexes in the WT and *prf1* mutant is discussed. The data presented in Figure 4e suggest lower stability of PSII supercomplexes in the *prf1* mutant (high abundance of the C2S form), which, in my opinion, is not consistent with the putative role of PRF1 in the degradation process. Can you clarify this part?
- Can you estimate a stoichiometry of alpha-tocopherolquinone per PSII monomer-TEF14-PRF1-PRF2 complex?

Other comments:

- Supplementary Video 1 shows variable conformations of the FAB domain of TEF14. Is this video based on structural data or just a simulation? Please clarify.
- Line 96: misspelling – muants
- Line 358: misspelling – cynobacterial

Reviewer #2:

Remarks to the Author:

General comments:

The manuscript presents the detailed structural and functional analysis of the unique intermediate complex during photosystem II (PSII) repair in the green alga *Chlamydomonas reinhardtii*. It is the first repair-specific PSII complex isolated and analysed up-to-now and from this point of view

the study deserves publication. The complex contains three auxiliary proteins. TEF14 is an orthologue of previously identified Arabidopsis luminal protein needed for monomerization of PSII(MPH2) during the repair, the other two small membrane proteins PRF1 and PRF2 are newly identified and described small proteins. Since PSII repair belongs to the key biochemical process maintaining the functionality of oxygenic photosynthesis and given the quality of data and their novelty, the manuscript deserves the publication in the journal. The study appears very well performed and provides a detailed mechanistic explanation of function of each identified protein: TEF14 displaces PsbO, PRF1 CP29 antenna and LHC, and PRF2 blocks the QB binding site. The work also revealed the tocopherol molecule bound in the vicinity of cyt b559 where it could act as electron acceptor from this cytochrome photoprotecting the complex. I have just a number of minor comments and suggestions which should be considered before the acceptance of the manuscript for publication.

Specific comments:

1. Page 5, line 96: mutants instead of muants
2. Page 5, line 98: there should be a short description what is PPH1 and why double instead of single pbcpc mutant was used for isolation of the complex
3. Page 6, lines 112-116 and Fig. 1b: PRF2 is practically invisible in the HL treated monomer of the pph1;pbcpc mutant, how can it be explained?
4. Fig. 1b: designations of lanes WT LL, WT HL, ppH1;pbcpc LL and ppH1;pbcpc HL of instead of existing ones would be more self-explaining. The designation of D1 is questionable, given the cleavage of the protein in HL treated cells of the mutant, the lower strong band does not appear to belong to D1. Instead, large amount of the described D1 fragments should be visible in the stained lane of HL treated mutant monomer: could the strong band below LhcMS bands represent the 18 kDa fragment?
5. Page 6, line 123: how did the authors judge that the monomers are partly damaged? Was their functional assay performed?
6. Page 7, line 135: DE loop is missing in the structure, is it due to its flexibility or is it really missing?
7. Fig 3: adding short headlines like TM HL and TEF14 TM HL to Figs. 3c and e, resp., would help them to become better self-explaining. It is not clear why in Fig. 3c the amount of PsbO in WT is so low. The difference between blots +TEF14 and -TEF14 in Fig. 3e is small and therefore needs to be well reproducible if should be considered a support for the proposed function of TEF14. In the legend to this figure the designation by red asterisks is not explained.
8. Fig 3 of extended data: choice of colours of CP47 and TEF14 residues is not optimal, they are rather similar, especially in Figs. D and e, one colour should be changed to be better distinguishable.
9. Page 6, lines 128-131: Are the proteins PsbJ, PsbO, PsbP, PsbQ, PsbR and PsbW present in larger PSII complexes separated by SDG or some of them are released and present just in the region of unassembled proteins?
10. Page 14, lanes 319-320: Is the presence of tocopherol present in the vicinity of cyt b559 exclusive for the repair PSII-TTP complex or can it be also detected by HPLC in the standard PSII? Can the authors confirm whether the tocopherol separated and detected by HPLC has also the corresponding absorption spectrum?
11. Page 16, line 357: Authors state that PRF1 plays a role different from the partly similar cyanobacterial Psb34. However, since cyanobacterial Hlips are ancestors of LHC proteins, are most probably bound to CP47 in similar way like CP29 and Psb34 prevents Hlip binding into this site, it seems more precise to conclude that the role of both proteins is analogous.
12. Page 16, line 375: There might be alternative scenarios here. For instance, the change of PSII conformation induced by photodamage may lead to PsbJ destabilization which is further facilitated by TRF2 binding and TRF2 would work here as an exchange factor for PsbJ. In that case PsbJ could be still present in the HL-treated PSII core of the PRF2 mutant and it would be worth to test it. Then the release of PsbJ upon TRF2 binding could be included into the model in Fig. 7.

Point-to-point responses to reviewer comments

Responses to Reviewer #1:

Li et al. revealed a cryo-EM structure of an intermediate complex (PSII-monomer-TEF14-PRF1-PRF2 complex) of the PSII repair cycle from *pph1;pbcp* *Chlamydomonas reinhardtii* mutant adapted to high-light stress. They also provided a biochemical and functional characterization of this complex, specifically of TEF14, PRF1, and PRF2. This characterization was supported by analysis of the generated knockout mutants and the purified recombinant TEF14 protein. In my opinion, the authors present a thorough study on the reparative processes of photosystem II. Thanks to the appropriately designed experimental approach, the presented description of the function of this complex is supported by the experiments performed. Thus, the described study will contribute significantly to our knowledge in the field of reparative processes of the photosynthetic apparatus, specifically photosystem II. In my opinion, the work is of great interest to a broad scientific community and the manuscript is suitable for a publication in *Nature Communications* after clarification of some points.

Response: We thank the reviewer for the supportive comments and insightful questions/suggestions.

Major comments:

(1) In the Fig 1b, the presence of PRF2 in the *pph1;pbcp* mutant is hardly visible compared to WT. Can you comment on that with respect to the presence of this subunit in the cryo-EM structure? Further, there are additional unassigned bands in both WT and *pph1;pbcp* mutant below the TEF14 band. Do you have any idea what these bands represents?

Response: During revision, we have repeated the SDS-PAGE analysis and stained the gel with Coomassie brilliant blue R250 (Response Fig. 1a), and the band of PRF2 appeared to be more obvious than the previous version. As the low-molecular-weight proteins are often poorly stained with Coomassie brilliant blue R250, the weak intensity of the PRF2 band may result from inadequate staining or low occupancy. In order to quantify the stoichiometry of PRF2 relative to the PSII-core subunit (such as CP47), we have stained the SDS-PAGE gel with the SYPRO Ruby dye in another repeat (Response Fig. 1b). SYPRO Ruby is a ruthenium complex-based fluorescence dye exhibiting high sensitivity and a broad linear dynamic range (1-2 ng to 1-2 μ g) for quantification of proteins on the SDS-PAGE gel (Lopez, M. F. *et al. Electrophoresis*, 21: 3673-3683, 2000). Through the method, the molar ratio of PRF2 relative to CP47 is estimated to be 0.31, while the PRF1:CP47 and TEF14:CP47 ratio are higher at 0.44 and 0.52, for the high-light adapted PSII core monomer sample. In comparison, the molar ratios of TEF14/CP47, PRF1/CP47 and PRF2/CP47 are much lower at ~ 0.1 in the low-light adapted PSII core monomer sample. The result indicates that only a

portion (instead of all) of PSII core complexes contains PRF2, PRF1 and TEF14, consistent with the cryo-EM data processing result as shown in Extended Data Fig. 2. Thereby, the PSII-TPP complex is enriched through both the biochemical and *in silico* purification processes. Note that the band of PRF2 is identified to be a mixture of PRF2 and a homologous protein (PRF2') according to the latest mass spectrometry and cryo-EM analysis results (see Response Figure 8 on p20 of this file, as well as line339-357, p9 and Extended Data Fig. 8 of the revised manuscript for more details).

Response Fig. 1: Protein composition analysis of the PSII-TPP complex. a, SDS-PAGE analysis of the PSII core complexes stained with the Coomassie brilliant blue R250. The protein identities of the bands detected by mass spectrometry were listed at the right side. The bands labeled as #1-6 contain various contaminant proteins as identified by mass spectrometry, namely Cyt *f* (#1), LhcbM proteins (#2), Lhca1 (#3), Cyt *b*₆ (#4), Psb27 (#5) and Lhca7 (#6). In the band corresponding to PRF2, it is mixed with PRF2' (Uniprot ID: A8HMM7) which is a newly identified protein sharing

sequence similarity with PRF2. **b**, The SDS-PAGE analysis of the PSII core complex samples prepared from high-light (HL) and low-light (LL) adapted cells, respectively. The gel was stained by the SYPRO Ruby dye and scanned on a Typhoon FLA 9500 imager. The relative amount of protein in the designated band is quantified as the fluorescence intensity of the band divided by the corresponding protein molecular weight. The molar ratios were calculated as the ratios between the relative amounts of the target protein (TEF14, PRF1 or PRF2) and that of CP47.

About the unassigned bands below TEF14 band, we have carried out protein identification through the mass spectrometry analysis on the proteolytic digestion products of the excised bands. As shown in Response Fig. 1a, the two major bands between TEF14 and PRF1 belong to fragments of D1/D2 and contaminant proteins (such as Psb27 and Lhca7 in the band labeled as #5, #6) respectively. For the low-molecular-weight bands below PRF2, they may belong to PsbE, PsbF and other small subunits and PsbK according to the recent mass spectrometry data and our previous work (Sheng X. et al. *Nat. Plants* 5: 1320-1330, 2019).

In the revised manuscript, we have replaced the original Fig. 1b with the improved version shown in Response Fig. 1a.

(2) The proposed model (Fig. 7) considers a binding competition between TEF14, PRF1, and PRF2 factors with other PSII core subunits, especially under high-light conditions. Is there any information on how the levels of these factors are controlled, e.g. under optimal light conditions, to reduce binding competition and avoid a disassembly of PSII supercomplexes when it is not necessary?

Response: This is a great question. We have analyzed the transcript level of TEF14, PRF1 and PRF2 by using the quantitative reverse transcription polymerase chain reaction (RT-qPCR) method (Response Fig. 2). The transcript levels of TEF14 and PRF2 are increased 1.5-fold after high-light ($330 \mu\text{mol photons m}^{-2} \text{s}^{-1}$) treatment for 4 hours (Response Fig. 2a and c). The transcript level of PRF1 remained almost unchanged during high-light treatment, except for a 50% decrease after 2 hours of high-light treatment (Response Fig. 2b). The results indicate that the expression levels of TEF14 and PRF2 are likely controlled by high-light irradiation, whereas the level of PRF1 may be less sensitive to high light (or even downregulated during the first 2 hours of high-light treatment). Thereby, disassembly of the PSII-LHCII supercomplexes may be avoided under low-light conditions, but facilitated under high-light conditions when TEF14 (and PRF2) are expressed at higher level and PRF1 could be recruited to the binding site nearby CP47 by TEF14 to prevent reassociation of CP29 with CP47. It is also noteworthy that TEF14 and PRF2 may contain multiple phosphorylation sites according to a previous phosphoproteome analysis of *C. reinhardtii* (Wang, H. et al. *Mol Cell Proteomics*, 13: 2337-2353, 2014), suggesting that they are likely regulated at both transcriptional and post-translational levels. The new results have been included in the revised manuscript as Extend Data Figure 1 and cited in the main text (lines 125-134, p4).

Response Fig. 2: The changes of the transcription levels of *TEF14*, *PRF1* or *PRF2* over time in response to the high-light treatment. a, *TEF14*; b, *PRF1*; c, *PRF2*. The light intensity for treating the *C. reinhardtii* cells is at $330 \mu\text{mol photons m}^{-2} \text{s}^{-1}$. The relative transcription levels were presented as mean \pm SD of three biological replicates (each containing three technical replicates). The transcription level at each time point was normalized to the level measured at 0 hour.

(3) In the Results part (lines 239-250), a role of PRF1 in disassembly of PSII supercomplexes in the WT and *prf1* mutant is discussed. The data presented in Figure 4e suggest lower stability of PSII supercomplexes in the *prf1* mutant (high abundance of the C₂S form), which, in my opinion, is not consistent with the putative role of PRF1 in the degradation process. Can you clarify this part?

Response: As the reviewer pointed out, the stability of PSII-supercomplex may appear to be lower in high-light adapted *prf1* mutant than WT, if we only consider the apparent increase of the C₂S form of PSII-LHCII supercomplex from 1.9% to 18.0%. Notably, according to the data shown in Fig. 4c, the abundance of the PSII-LHCII supercomplex in *prf1* mutant cells is approximately 6-7 times of that in WT (as estimated by the peak area). Therefore, the intact PSII-LHCII complexes (C₂S₂ and larger ones) and the partial ones (C₂S) are both more abundant in the *prf1* mutant than in the WT.

Firstly, the increased abundance of C₂S form of PSII-LHCII supercomplex in the *prf1* mutant strain may suggest that the C₂S₂ or larger PSII-LHCII supercomplexes are either less stable in the *prf1* mutant (as the reviewer suggested), or have loose peripheral antenna complexes readily dissociated from the PSII core of C₂S₂ without the assistance of PRF1 (or facilitated by some other unknown factors instead). Secondly, as the binding site of PRF1 is located on the surface of CP47 partially overlapping with that of CP29 (Fig. 4a), it may either compete with CP29 for the binding site or prevent CP29 from binding to CP47 after it binds to CP47. The lack of PRF1 in the mutant may result in less conversion of C₂S to C₂ and less conversion of C₂S₂ to C₂S during the disassembly process, when dissociation of CP29 (as well as LHCII and CP26 subsequently) from the PSII-LHCII supercomplexes is blocked or impeded. On the other hand, as the binding site of PRF1 is distant from those of the moderately-associated LHCII (M-LHCII) and loosely-associated LHCII (L-LHCII), the conversion

of C₂S₂M, C₂S₂M₂, C₂S₂ML and C₂S₂M₂L₂ to C₂S₂, or the conversion of C₂SML to C₂S during the disassembly process is most likely independent on the presence/absence of PRF1. Thirdly, it is also noteworthy that the C₂S supercomplexes (at least in part) are probably also the products of the *de novo* assembly of PSII with C₂ assembled with CP29, S-LHCII and CP26 on one side. As a result, there appear to be more C₂S₂ and C₂S-type supercomplexes in the *prf1* mutant than the WT.

Alternatively, the PSII-LHCII complexes in the *prf1* mutant may suffer from more severe photo-oxidative damage under high-light conditions, as CP29, CP26 and S-LHCII in the *prf1* mutant cannot dissociate from PSII efficiently. Previously, Kale et al. (*Photosynthesis Research*, 152: 261-274, 2022) identified multiple ROS modification sites in LHCII, including Pro164 from Lhcb1 located at the LHCII-CP29 and/or LHCII-CP26 interfaces. As the antenna dissociation step during the PSII repair process is likely blocked in the *prf1* mutant, the accumulation of excess excitation energy in the antenna domains may lead to oxidative modifications of the LHCII residues (in addition to the damages in the PSII core). The effect may further render the peripheral antennae unstable and resulting in the higher ratio of C₂S complex (due to a disassembly process independent of PRF1), as observed by the negative-staining electron microscopy results. We have revised the manuscript and included the possibility of lower stability of PSII-LHCII supercomplexes in the *prf1* mutant (lines 270-278 in p7).

(4) Can you estimate a stoichiometry of alpha-tocopherolquinone per PSII monomer-TEF14-PRF1-PRF2 complex?

Response: To address the question and estimate the stoichiometry of α -tocopherol quinone in the PSII-TPP sample, we extracted the organic ligand molecules from the sample by using n-hexane, and carried out HPLC analysis on the extract with the eluted fractions monitored by a photo-diode array (PDA) detector. As shown in Response Fig. 3a, α -TQ is present in the extracted sample along with several other PSII ligands, including chlorophyll *a* (Chl *a*), β -carotene and plastoquinone (PQ). To estimate the stoichiometry of α -TQ per PSII-TPP complex, β -carotene is chosen as a relatively specific internal reference for estimating the amount of PSII for three reasons: 1) The β -carotene molecules are mainly associates with PSII core at the peripheral regions like α -TQ (Response Fig. 3b). Therefore, the accessibility of the two molecules to the organic solvent (n-hexane) should be similar; 2) Both α -TQ and β -carotene are highly soluble in n-hexane (Kruk, *Biophysical Chemistry*, 32: 51-56; Ashenafi et al., *Food Chemistry Advances* 2: 100178). 3) The PSII-TPP sample prepared in this work is contaminated by LHCII (with the apoproteins of LhcbMs) and Cyt *b₆f* (Fig. 1b and Extended Data Fig. 2a). While LHCII and Cyt *b₆f* both contains Chl *a*, they either do not have β -carotene (LHCII) or only have two carotene molecules (Cyt *b₆f* dimer) associated with them. The maximum absorption wavelength of α -TQ (268 nm) was used to monitor the compounds eluted from the HPLC column. The molar extinction coefficient (ϵ) of α -TQ and β -carotene are 18.2 mM⁻¹ cm⁻¹ (Gille et al., *Biochem.*

Pharmacol. 68(2):373-81) and $20.8 \text{ mM}^{-1} \text{ cm}^{-1}$ (converted from Zechmeister et al., J. Am. Chem. Soc. 65, 1522-1528), respectively. The molar amount (in relative unit) of α -TQ or β -carotene in the sample was estimated by dividing the absorption peak area by the corresponding ϵ value (according to the Beer's law expressed as $c = A/\epsilon L$; c , concentration; A , absorbance; ϵ , molar extinction coefficient; L , light path length in centimeters). Considering that there are ten β -carotene molecules associate with one PSII-TPP complex, the molar ratio of α -TQ to PSII-TPP is calculated as the molar amount of α -TQ divided by one-tenth of the molar amount of β -carotene (Response Fig. 3c). The molar ratio of α -TQ vs the PSII-TPP complex in the sample is about 0.23:1 according to the above estimation. The result suggests that only a small portion of the PSII core complexes in the sample contains α -TQ. During cryo-EM data processing, the particles with α -TQ (and TEF14, PRF1 and PRF2) associated have been enriched through multiple rounds of classification process (Extended Data Fig. 2). It is also noteworthy that similar amount of α -TQ also exists in the high-light adapted PSII-LHCII supercomplexes, but the content in the low-light PSII-LHCII is much lower (only around one-fifth of that observed in the high-light PSII complexes, Response Figure 3c). The new HPLC data have been included as Fig. 6f and g, and the stoichiometry of α -TQ per PSII-M is described in the main text of the revised manuscript (lines 377-382, p10).

Response Fig. 3: Estimation of the stoichiometry of α -TQ within the PSII complexes. a, HPLC separation of the ligands extracted from the PSII samples

followed by the photo-diode array (PDA) detection. All chromatographic traces were normalized basing on the absorption peak value of β -carotene. A zoom-in view of the absorption peaks corresponding to α -TQ is shown at the upper left inset panel. The characteristic absorption spectrum of α -TQ from the high-light PSII-TPP sample as detected by the PDA monitor was listed at the upper right panel. **b**, The structural model of the PSII-TPP complex viewed from the stromal side showing the relative location of the α -TQ and β -carotene molecules. Both of the two molecules associate with the peripheral regions of PSII core. **c**, Summary of the data on the molar ratio of α -TQ per PSII core monomer based on the HPLC profiles of the ligands extracted by n-hexane from three different PSII samples.

(5) Other comments:

Supplementary Video 1 shows variable conformations of the FAB domain of TEF14. Is this video based on structural data or just a simulation? Please clarify.

Response: The video is a morph movie generated by using the structural models fitted in the corresponding cryo-EM maps of two different 3D classes with TEF14 trapped in distinct conformations. We have included the detailed description in the revised SI Video caption as shown in the supplementary file (p6).

Line 96: misspelling – muants; Line 358: misspelling – cynobacterial

Response: Thanks for pointing out the typo errors. We have corrected them in the revised manuscript.

Responses to Reviewer #2

General comments:

The manuscript presents the detailed structural and functional analysis of the unique intermediate complex during photosystem II (PSII) repair in the green alga *Chlamydomonas reinhardtii*. It is the first repair-specific PSII complex isolated and analysed up-to-now and from this point of view the study deserves publication. The complex contains three auxiliary proteins. TEF14 is an orthologue of previously identified *Arabidopsis* luminal protein needed for monomerization of PSII(MPH2) during the repair, the other two small membrane proteins PRF1 and PRF2 are newly identified and described small proteins. Since PSII repair belongs to the key biochemical process maintaining the functionality of oxygenic photosynthesis and given the quality of data and their novelty, the manuscript deserves the publication in the journal. The study appears very well performed and provides a detailed mechanistic explanation of function of each identified protein: TEF14 displaces PsbO, PRF1 CP29 antenna and LHC, and PRF2 blocks the QB binding site. The work also revealed the tocopherol molecule bound in the vicinity of cyt b559 where it could act as electron acceptor from this cytochrome photoprotecting the complex. I have just a number of minor comments and suggestions which should be considered before the acceptance of the manuscript for publication.

Response: We thank the reviewer for very helpful suggestions and insightful comments.

Specific comments:

1. Page 5, line 96: mutants instead of muants

Response: We have corrected the typo and the other ones in the revised manuscript.

2. Page 5, line 98: there should be a short description what is PPH1 and why double instead of single pbcp mutant was used for isolation of the complex.

Response: Thanks a lot for the advice. PPH1 is a chloroplast metal-dependent protein phosphate originally identified in in *Arabidopsis thaliana* as a crucial enzyme involved in state transitions (Shapiguzov, A. et al. *Proc. Natl. Acad. Sci USA* 107: 4782-4787, 2010; also named as TAP38 by Pribil et al. *Plos Biol.* 8: e1000288, 2010). It mainly functions to dephosphorylate LHCI during state transitions (Shapiguzov, A. et al. *Proc. Natl. Acad. Sci USA*, 2010; Pribil et al. *Plos Biol.* 2010; Wei, X. P. et al. *Plant Cell* 27: 1113–1127, 2015). Moreover, PPH1 may also dephosphorylate PSII core proteins when it is overexpressed in plant cells (Pribil et al. *Plos Biol.* 2010) or incubated with thylakoid membrane under *in vitro* conditions (Wei, X. P. et al. *Plant Cell* 2015). Previous studies reported that dephosphorylation of PSII core subunit is a prerequisite for degradation of the damaged D1 protein (Rintamäki et al. *J. Biol. Chem.* 271: 14870-5, 1996 and Puthiyaveetil et al. *Plant Cell Physiol.* 55: 1245-1254, 2014). In

Chlamydomonas reinhardtii, PBCP and PPH1 may have distinct but overlapping sets of targets (Cariti, F. et al. *Plant Physiol.* 183: 1749-1764, 2020). In the single *pbcp* mutant, the presence of PPH1 might still dephosphorylate PSII core subunits at a lower rate than PBCP. Therefore, we decided to use the *pph1;pbcp* (the double phosphatase mutant strain) for the experiment of high-light treatment ($330 \mu\text{mol photons m}^{-2} \text{s}^{-1}$) so as to capture the PSII core complexes arrested at the intermediate state (before the D1 proteins are fully degraded) as much as possible. We have included the above description in the revised manuscript (lines 93-104, p3).

3. Page 6, lines 112-116 and Fig. 1b: PRF2 is practically invisible in the HL treated monomer of the *pph1;pbcp* mutant, how can it be explained?

Response: The weak band intensity of PRF2 may be due to poor resolution of the low-molecule-weight bands, insufficient staining by the Coomassie brilliant blue R250 dye or over-destaining process, which are common problems for the low-molecular-weight protein bands on the SDS-PAGE gel. To solve the problem, we have repeated the experiment during revision and obtained an improved gel image (see Response Fig. 1a shown below). The band of PRF2 appears stronger, as shown by the images of SDS-PAGE gel stained by the Coomassie brilliant blue R250 (Response Fig. 1a). Moreover, we have used the SYPRO Ruby dye (a ruthenium complex-based fluorescence dye exhibiting high sensitivity and a broad linear dynamic range 1-2 ng to 1-2 μg ; Lopez, M. F. et al. *Electrophoresis*, 21: 3673-3683, 2000) to stain the SDS-PAGE gel for quantification of the proteins in the target bands (Response Fig. 1b). The result suggests that TEF14, PRF1 and PRF2 may associate with PSII core at a sub-stoichiometric level (0.52, 0.44 and 0.31 molecule per CP47 or per PSII core monomer, respectively) in the sample from the high-light adapted cells (Response Fig. 1b, the bottom table, HL). Such levels are much higher than the corresponding ones in the PSII core sample from the low-light adapted cell (LL). Note that the PSII-TPP complex particles with TEF14, PRF1 and PRF2 associated were further separated from those without them and enriched *in silico* through multiple rounds of 3D classification process as shown in Extended Data Fig. 2.

Response Fig. 1: Protein composition analysis of the PSII-TPP complex. a, SDS-PAGE analysis of the PSII core complexes stained by the Coomassie brilliant blue R250. The protein identities of the bands detected by mass spectrometry were listed at the right side. The bands labeled as #1-6 contain different contaminant proteins as identified by mass spectrometry, namely Cyt *f* (#1), LhcbM proteins (#2), Lhca1 (#3), Cyt *b*₆ (#4), Psb27 (#5) and Lhca7 (#6). **b**, SDS-PAGE analysis of the PSII core complex samples prepared from high-light (HL) and low-light (LL) adapted cells, respectively. The gel was stained by the SYPRO Ruby dye and scanned on a Typhoon FLA 9500 imager. The relative amount (in arbitrary unit) of protein in the designated band is quantified as the fluorescence intensity of the band divided by the corresponding protein molecular weight. The molar ratios were calculated as the ratios between the relative amounts of the target protein (TEF14, PRF1 or PRF2) and that of CP47.

4. Fig. 1b: designations of lanes WT LL, WT HL, ppH1;pbcp LL and ppH1;pbcp HL of instead of existing ones would be more self-explaining. The designation of D1 is questionable, given the cleavage of the protein in HL treated cells of the mutant, the lower strong band does not appear to belong to D1. Instead, large amount of the described D1 fragments should be visible in the stained lane of HL treated mutant monomer: could the strong band below LhcMS bands represent the 18 kDa fragment?

Response: We have updated the labels in revised version of Fig. 1b accordingly. About the designation of D1 protein, the 25 kDa band we previously assigned as D1 protein in the high-light adapted PSII-M sample from *pph1;pbcp* contains LHCII proteins (LhcbMs) in addition to the D1 protein (PsbA), according to the Orbitrap mass spectrometry analysis result (Response Figure 4a, Band 2). It is likely that the D1 protein contained in this band may be a large fragment of D1 protein. To search for the D1 protein, we have carried out MALDI-TOF mass spectrometry analysis on the higher-molecular-weight band (Band 1) above Band 2. The result demonstrates that Band 1 mainly contains the D1 protein (PsbA, Response Figure 4a). Below Band 2, the 18 kDa band (Band 3) contains Lhca1 and Cyt *b*₆ (Response Fig. 4a and c). They are contaminant proteins from the thylakoid membrane. In order to get rid of those contaminants, we applied a second sucrose density gradient (SDG) for the purified PSII-TPP complex (Response Fig. 4c). The SDS-PAGE and mass spectrometry results showed that the 18 kDa protein band in the PSII-TPP fraction from the second SDG tube still mainly contain Lhca proteins (Lhca5, Lhca3 and Lhca4). It is possible that the Lhca proteins co-migrating with PSII-TPP complex on the SDG may come from a “LHCI-belt” subcomplex composed of eight Lhca monomers dissociated from the PSI-LHCI supercomplex. The LHCI-belt subcomplex has a molecular weight of about 300 kDa, which is close to that of the PSII-TPP complex (~ 310 kDa). While it is likely that the band below 20 kDa marker (or below the “LhcbMs” bands) may contain the 18 kDa fragment of D1 protein (as demonstrated in the western blot shown in Extended Data Figure 3d), no D1 fragments of that size could be detected in the SDS-PAGE gel bands, presumably due to the presence of LHCI contaminants and the resistance of the D1 fragment to further proteolytic digestion treatment for mass spectrometry analysis (a common problem for membrane proteins).

Response Fig. 4: Search for the D1 protein bands on the SDS-PAGE gel. a, Protein identification through the mass spectrometry analysis on the proteolytic digestion products of the three excised bands from the SDS-PAGE gel. The SDS-PAGE gel figure was taken from Fig. 1b in the previous manuscript. Band 2 was previously assigned as D1, and is now corrected as LhcbM protein contaminants according to the mass spectrometry data (shown on the right side). Band 3 is the band at around 18 kDa position suspected to be a D1 fragment, and is now identified as Lhca1 and Cyt *b*₆. Band 1 mainly contains the D1 protein (PsaA). The mass spectrometry results were listed on the right side. **b,** In order to remove the contaminant proteins, a second round of sucrose density gradient ultracentrifugation was employed using the PSII-TPP complex sample as input. The result of SDS-PAGE analysis and the following mass spectrometry identification results are listed on the right side. The low-light PSII-M and

LHCII samples are loaded as references for the bands of the major PSII subunits (CP47, CP43, D2 and D1) and LhcbMs respectively.

5. Page 6, line 123: how did the authors judge that the monomers are partly damaged? Was their functional assay performed?

Response: As revealed by the cryo-EM map, the Mn_4CaO_5 cluster in the PSII-TPP complex is obviously absent and the DE-loop region of D1 appears to be partly degraded. Basing on the observations, we reasoned that our PSII-TPP complex is partly damaged. To further verify it, we have analyzed the function of PSII-TPP by using the oxygen evolution assay and the F_v/F_m (indicator of the maximum quantum yield of PSII chemistry) measurement, respectively (Response Fig. 5). The low-light adapted PSII-LHCII supercomplex sample from the *pbcp;pph1* mutant is active in generating oxygen with an oxygen-evolving rate at $65 \mu\text{mol O}_2 \text{ mg}^{-1} \text{ Chl h}^{-1}$ (it may reach $210\text{-}270 \mu\text{mol O}_2 \text{ mg}^{-1} \text{ Chl h}^{-1}$ under optimal conditions for the supercomplex from WT; Sheng X. et al. *Nat. Plants* 5: 1320–1330, 2019), whereas the oxygen evolving activity was barely detectable for the PSII-TPP sample upon illumination with saturate actinic light (Response Fig. 5a). Besides, the F_v/F_m value for the low-light PSII-LHCII sample purified from the *pbcp;pph1* mutant is 0.38, whereas it is close to zero for the PSII-TPP sample (Response Fig. 5b). The results collectively support that the PSII-TPP complex is inactive at both electron donor and acceptor side, due to the damages on the oxygen-evolving center and the D1 subunit.

Response Fig. 5: Functional assays on the PSII-TPP complex sample. a, Oxygen evolving activity assay of the PSII-TPP complex sample. The PSII-LHCII complex prepared from low-light adapted cells were also tested as a positive control. The figure is a representative data from a set of three technical replicates. **b,** Maximal quantum efficiency measurement of PSII. The F_v/F_m values are presented as mean \pm SD of three technical replicates.

6. Page 7, line 135: DE loop is missing in the structure, is it due to its flexibility or is it really missing?

Response: According to our western blot analysis (Extended Data Figure 3d), the D1 protein in the PSII-TPP complex is likely cleaved at the DE-loop region with at least one cleavage site. As shown in our recent SDS-PAGE result (Response Fig. 1a), the amount of full-length D1 protein is much lower in the high-light treated samples than in the low-light adapted samples, supporting that a portion of the D1 protein may have been cleaved by proteases. After losing restraints from the covalent peptide bond, the loop region may become more flexible and therefore appears to be invisible in the cryo-EM map (which is an averaged reconstruction from thousands of particles). On the other hand, it is also possible that the entire loop region is cleaved off, which will also lead to missing density in the local region of the cryo-EM map if the peptide fragment is released from PSII core. Nevertheless, due to the current technical limitations, it is not possible to distinguish the two different situations. We have rewritten the statement in the revised manuscript (lines 160-162, p5).

7. Fig 3: adding short headlines like TM HL and TEF14 TM HL to Figs. 3c and e, resp., would help them to become better self-explaining. It is not clear why in Fig. 3c the amount of PsbO in WT is so low. The difference between blots +TEF14 and -TEF14 in Fig. 3e is small and therefore needs to be well reproducible if should be considered a support for the proposed function of TEF14. In the legend to this figure the designation by red asterisks is not explained.

Response: We thank the reviewer for the helpful advices and comments. As the reviewer suggested, we have updated the labels in Figs. 3c and e (updated as Figs. 3 and f in the revised version). About the question concerning the apparent low amount of PsbO in the WT strain, the previous western blots from WT and *tef14* strain were photographed separately, and the exposure time was determined by the imaging machine automatically. As a result, the intensities between WT and the *tef14* strain are likely at different scale. In order to quantify the level of PsbO associated with PSII more precisely, we collected the PSII-M/D sample from the SDG tubes (as in Fig. 3c), and analyzed the relative amount of PsbO versus CP47 (a major core subunit of PSII core) using western blots. The blot intensity of CP47 was used as an internal reference to calibrate the amount of PSII core (Response Fig. 6a). Remarkably, PsbO is present at in the PSII-D sample from the *tef14* mutant at a level about twice as much as the one from WT. Meanwhile, the level of PsbO associated with PSII-M does not exhibit significant difference between the WT and *tef14* strains. This is in line with the structural observation that the binding site of TEF14 has some clashes with that of PsbO' from the adjacent monomer of the PSII dimer (Fig. 3b).

We also carried out similar analysis for the detergent-solubilized *tef14* thylakoid, with or without the purified recombinant TEF14 protein, to check whether TEF14 can facilitate the dissociation of PsbO from PSII. The PSII-M and PSII-D samples were purified and collected from *tef14* mutant thylakoid for the western blot analysis against

CP47 and PsbO respectively (Response Fig. 6b). A slight decrease of PsbO level was detected in the PSII-D sample with exogenous TEF14 protein (relative to the one without the exogenous TEF14 protein), whereas there was no significant change in the PSII-M sample w/o TEF14 added. Therefore, the above results collectively demonstrate that TEF14 is likely involved in facilitating the dissociation of PsbO from PSII-D. The effect of adding exogenous TEF14 protein is not as strong as the endogenous one (PSII-D of WT vs *tef14* in response Fig. 6a), suggesting that there might be other factors cooperating with TEF14 in the PsbO dissociation process *in vivo*. For instance, it was previously reported that a thylakoid luminal protease named HhoA is responsible for degradation of PsbO in a thioredoxin-dependent way (Roberts et al. Plos one. 7: e45713, 2012). In the revised manuscript, Figure 3 has been updated to include the new data and the related statements have also been updated (lines 199-202, 209-216, p5-6).

Response Fig. 6: TEF14 is involved in facilitating the dissociation of PsbO from PSII. **a**, The PSII-M and PSII-D samples were purified from WT and *tef14* thylakoid, and subjected to western blot analysis by using the antibodies against CP47 and PsbO. The intensity of the western blots against CP47 was used as an internal reference to calibrate the amount of PSII core. The blot intensity of PsbO divided by that of CP47 was calculated and plotted on the right side. The highest value of PsbO/CP47 was normalized to 1 for clarity. The PsbO/CP47 values are represented as mean \pm SD of three technical replicates. **b**, The PSII-M and PSII-D samples were purified from *tef14* thylakoid with or without the addition of recombinant TEF14 protein, and was subjected to western blot analysis with the antibodies against CP47 and PsbO. The normalized PsbO/CP47 blot intensity is plotted on the right side. The PsbO/CP47 values are represented as mean \pm SD of three technical replicates. The western blots of PsbO and CP47 from the same gel are imaged in two separate batches (one batch for PsbO and the other for CP47) due to the capacity limit of the imager.

8. Fig 3 of extended data: choice of colours of CP47 and TEF14 residues is not optimal, they are rather similar, especially in Figs. D and e, one colour should be changed to be better distinguishable.

Response: We have adjusted the colors of CP47 and TEF14 in Extended Data Figure 3 (or ED Figure 4 in the revised manuscript) to enhance the color contrast and distinguish TEF14 better from CP47.

9. Page 6, lines 128-131: Are the proteins PsbJ, PsbO, PsbP, PsbQ, PsbR and PsbW present in larger PSII complexes separated by SDG or some of them are released and present just in the region of unassembled proteins?

Response: To address the question, we have collected a new cryo-EM dataset for the PSII-LHCII supercomplex purified from the high-light adapted *ppl1;pbc* double mutant cell, and obtained the cryo-EM map refined to an overall resolution of 3.2 Å. As shown in Response Figure 7, PsbO and PsbW are present in the supercomplex, whereas PsbJ, PsbP, PsbQ and PsbR are likely missing as their densities are too weak to be observed in the cryo-EM map. The results indicated that PsbJ, PsbP, PsbQ and PsbR may be released at the early stage of PSII repair process, whereas PsbO and PsbW are released at a later stage. As mentioned above, PsbO dissociation may be facilitated by TEF14 binding. PsbW is a small intrinsic protein sandwiched between the strongly-associated LHCII and PSII core (Wei, X. P. *et al. Nature* 534: 69-74, 2016; Sheng X. *et al. Nat. Plants* 5: 1320-1330, 2019). It is probably released during or after the disassembly of peripheral antenna complexes from PSII.

Response Fig. 7: The cryo-EM densities for the high-light adapted PSII-LHCII supercomplex from the *ppl1;pbc* mutant. The structural model for the low-light adapted PSII-LHCII supercomplex (PDB: 6KAC, shown as ribbons) was fitted into the map of the high-light adapted PSII-LHCII supercomplex (shown as grey surface). The models fit well with the density except for those of PsbJ, PsbR, PsbP and PsbQ (highlighted in magenta). Shown on the right are the densities of PsbO and PsbW fit with the corresponding structural models.

10. Page 14, lanes 319-320: Is the presence of tocopherol present in the vicinity of cyt b559 exclusive for the repair PSII-TPP complex or can it be also detected by

HPLC in the standard PSII? Can the authors confirm whether the tocopherol separated and detected by HPLC has also the corresponding absorption spectrum?

Response: To obtain the answers to the question, we have separated the organic ligands (extracted with n-hexane) associated with the PSII-TPP complex and the standard PSII (i.e. PSII-LHCII supercomplex) through the HPLC column and measured the absorption spectra of the eluted fraction by using a photo-diode array (PDA) detector. As shown in Response Fig. 3a, the α -tocopherol quinone fraction extracted from the PSII-TPP complex sample does have a characteristic absorption spectrum with a broad characteristic peak ranging from 262 to 268 nm matching with those of a standard α -tocopherol quinone as reported in the previous publications (e.g. Harrison et al. *Biochim Biophys Acta*. 21: 150-158, 1956; see also Fig. 6f in the revised manuscript).

Besides, we have also estimated the relative amount of α -TQ in the low-light and high-light adapted PSII-LHCII supercomplexes (Response Fig. 3a and c). The abundance of α -TQ in the high-light PSII-LHCII supercomplex is similar to (or slightly higher than) the one in the PSII-TPP sample, whereas the low-light PSII-LHCII supercomplex has a much lower level of α -TQ in comparison with the high-light PSII samples. Therefore, the level of α -TQ may be increased under high-light conditions and can associate with both the PSII-TPP complex and the PSII-LHCII supercomplex. Recently, an α -TQ or α -tocopherol (α -Toc) molecule has been located at the interface between the LHCII trimer and CP43 in the PSII-LHCII supercomplex from Spruce (Opatíková., M. *et al. Nat. Plants* 9: 1359-1369, 2023). The binding site is different from the one we observed in the PSII-TPP complex.

The HPLC profiles, the absorption spectrum of the α -TQ fraction and the related description have been included in the revised manuscript (Fig. 6f and g, lines 377-382, p10).

Response Fig. 3: Estimation of the stoichiometry of α -TQ within different PSII complexes. **a**, HPLC separation of the small molecules extracted by n-hexane from the different PSII complexes followed by the photo-diode array (PDA) detection. All chromatographic traces were normalized based on the absorption peak value of β -carotene. A zoom-in view of the absorption peaks corresponding to α -TQ is shown on the upper left side. The characteristic absorption spectrum of α -TQ from the high-light PSII-TPP sample as measured by the PDA detector is shown on the upper right side. PSII-SC (LL) and PSII-SC (HL) are the PSII-LHCII supercomplex samples purified from low-light and high-light adapted cells respectively. **b**, A surface structural model of the PSII-TPP complex viewed from the stromal side showing the relative location of the α -TQ and β -carotene molecules. Both are associated with PSII core at the peripheral regions. **c**, A summary of the molar ratio of α -TQ per PSII core monomer in three different PSII samples.

11. Page 16, line 357: Authors state that PRF1 plays a role different from the partly similar cyanobacterial Psb34. However, since cyanobacterial Hlips are ancestors of LHC proteins, are most probably bound to CP47 in similar way like CP29 and Psb34 prevents Hlip binding into this site, it seems more precise to conclude that the role of both proteins is analogous.

Response: Thanks a lot for the insightful comment. We have discussed in the revised manuscript about the potential roles of PRF1 and Psb34 as analogous proteins in terms of their putative functions in preventing CP29/Hlip from binding to CP47 during the PSII repair cycle (lines 417-423, p11).

12. Page 16, line 375: There might be alternative scenarios here. For instance, the change of PSII conformation induced by photodamage may lead to PsbJ destabilization which is further facilitated by PRF2 binding and PRF2 would work here as an exchange factor for PsbJ. In that case PsbJ could be still present in the HL-treated PSII core of the PRF2 mutant and it would be worth to test it. Then the release of PsbJ upon PRF2 binding could be included into the model in Fig. 7.

Response: This is a good point and thanks for the helpful suggestion. To analyze whether PsbJ is present in the HL-treated PSII core of the *prf2* mutant, we have collected the cryo-EM data for the monomeric PSII core sample prepared from high-light adapted *prf2* mutant cells and solved the structure at 2.9 Å resolution. Evidently, PsbJ is also absent in this structure (Response Fig. 8a). As also mentioned above, the PSII-LHCII supercomplex from the high-light adapted cells also lacks PsbJ (Response Fig. 7). Therefore, PsbJ is most likely released at an early stage of the PSII repair process before PRF2 binds to PSII core as discussed previously.

To our surprise, the HL-treated PSII-M from the *prf2* mutant contains a new protein with a single transmembrane helix occupying the binding site of PRF2 (Response Fig. 8a). As the *PRF2* gene has already been knocked out successfully in this mutant, the density likely belongs to a new protein similar to PRF2. We went through the mass spectrometry results looking for other small membrane proteins in the “PRF2 band” from the PSII-TPP complex sample prepared from the *pph1;pbcp* mutant cells (Response Fig. 1a). Strikingly, a new small membrane protein (Uniprot: A8HMM7, Response Fig. 8b) is present in the band along with PRF2 with a high score. The model of this protein matches well with the cryo-EM density of the corresponding helix in the PRF2-depleted PSII-M complex (Response Fig. 8c). Hence, we have named this newly-identified protein subunit as PRF2', because its amino acid sequence is similar to that of PRF2, especially in the transmembrane-helix region (Response Fig. 8d). Nevertheless, the cryo-EM density of PRF2' is slightly different from that of PRF2 at the site interacting with W35_{PsbE} (Response Fig. 8e). Considering that PRF2 and PRF2' are both detected in the PSII-TPP sample by mass spectrometry, and that their structures and binding sites are similar, it is likely that the cryo-EM density previously assigned as PRF2 may be a mixture composed of both PRF2 and PRF2'. PRF2' may serve as a redundant homolog of PRF2 functioning in a similar way during the PSII repair process.

Response Fig. 8: Analyzing the binding sites of PsbJ and PRF2 in the monomeric PSII core complex purified from the high-light adapted *prf2* mutant cells. a, The cryo-EM map of the PSII-TPP complex from the high-light adapted *prf2* mutant (left) in comparison with the one from the *pph1;pbcp* mutant (right). The vacant PsbJ sites are indicated by the yellow dashed boxes. The densities of PRF2 and a similar protein named PRF2' are highlighted in magenta and purple respectively, while the remaining parts are in grey. **b**, Mass spectrometry results from two repeats indicate that the 7-kDa band excised from the SDS-PAGE gel shown in Fig. 1b contains both PRF2 and PRF2'. The other proteins in the list are likely from contaminants. **c**, Superposition of the PRF2' structural model with the local cryo-EM density map. **d**, Alignment of the amino acid sequences of PRF2 and PRF2'. The transmembrane helix regions of PRF2 and PRF2' are represented as magenta and purple tubes above the sequences, respectively. The highly similar regions within the transmembrane helices are framed in the black rectangle. **e**, The local characteristic cryo-EM density features distinguishing PRF2 from PRF2'. The structural models are superposed with the local cryo-EM densities.

It is also noteworthy that PRF2' was previously named as a putatively Photosystem B Associated 1 (PBA1), and was discovered to associate mainly with PSII monomer (Spaniol et al. *J Exp Bot.* 73: 245–262, 2022), consistent with our observation. Our result indicates that PBA1 (or PRF2') associates with PSII-M in a manner similar to that of PRF2, and may have similar function in inhibiting electron transport at the acceptor side of PSII during PSII repair. For future study, it will be interesting to generate and analyze the phenotypes of the mutant with *prf2'* knocked out and the *prf2;prf2'* double mutant.

The new results about PRF2' have been included as Extended Data Figure 8 and described in lines 340-358, p9 of the revised manuscript.

Reviewers' Comments:

Reviewer #1:

Remarks to the Author:

In my opinion, the authors of the manuscript have answered all my questions and made adequate corrections in the text of the manuscript. Therefore, I have no further questions or objections to the acceptance of the manuscript.

Reviewer #2:

Remarks to the Author:

General comments:

The manuscript has been amended according to the reviewers' comments and specifically concerning my comments, they all were seriously considered and most of them were also reflected in the manuscript modifications. To justify the response and corresponding modifications of the manuscript, a number of additional confirmatory experiments and analyses were performed and some additional new data were obtained (e.g. a PRF2 homologue PRF2' was identified in the structure of the repair complex). Thus, although I noticed two errors mentioned below, after their correction and additional double-checking I recommend the revised manuscript for publication in the journal.

Errors:

1. Line 1379: there should be Arg 225 instead of Arg255
2. Line 768: there should be "associated" instead of "associate"

Point-to-point Responses to the reviewers' comments

Reviewer #1

In my opinion, the authors of the manuscript have answered all my questions and made adequate corrections in the text of the manuscript. Therefore, I have no further questions or objections to the acceptance of the manuscript.

Response: We thank the referee for reviewing our work and the positive feedbacks.

Reviewer #2

General comments:

The manuscript has been amended according to the reviewers' comments and specifically concerning my comments, they all were seriously considered and most of them were also reflected in the manuscript modifications. To justify the response and corresponding modifications of the manuscript, a number of additional confirmatory experiments and analyses were performed and some additional new data were obtained (e.g. a PRF2 homologue PRF2' was identified in the structure of the repair complex). Thus, although I noticed two errors mentioned below, after their correction and additional double-checking I recommend the revised manuscript for publication in the journal.

Response: We thank the referee for reviewing our manuscript carefully and the supportive comment.

Errors:

- 1. Line 1379: there should be Arg 225 instead of Arg255**
- 2. Line 768: there should be "associated" instead of "associate"**

Response: Thank you very much for pointing out the typo errors. They have been corrected in the revised manuscript.